# FASTER SAMPLING FROM LOG-CONCAVE DENSITIES OVER POLYTOPES VIA EFFICIENT LINEAR SOLVERS

**Oren Mangoubi**
Worcester Polytechnic Institute

**Nisheeth K. Vishnoi**
Yale University

## ABSTRACT

We consider the problem of sampling from a log-concave distribution $\pi(\theta) \propto e^{-f(\theta)}$ constrained to a polytope $K := \{\theta \in \mathbb{R}^d : A\theta \leq b\}$, where $A \in \mathbb{R}^{m \times d}$ and $b \in \mathbb{R}^m$. The fastest-known algorithm Mangoubi & Vishnoi (2022) for the setting when $f$ is $O(1)$-Lipschitz or $O(1)$-smooth runs in roughly $O(md \times md^{\omega-1})$ arithmetic operations, where the $md^{\omega-1}$ term arises because each Markov chain step requires computing a matrix inversion and determinant (here $\omega \approx 2.37$ is the matrix multiplication constant). We present a nearly-optimal implementation of this Markov chain with per-step complexity which is roughly the number of non-zero entries of $A$ while the number of Markov chain steps remains the same. The key technical ingredients are 1) to show that the matrices that arise in this Dikin walk change slowly, 2) to deploy efficient linear solvers that can leverage this slow change to speed up matrix inversion by using information computed in previous steps, and 3) to speed up the computation of the determinantal term in the Metropolis filter step via a randomized Taylor series-based estimator. This result directly improves the runtime for applications that involve sampling from Gibbs distributions constrained to polytopes that arise in Bayesian statistics and private optimization.

## 1 INTRODUCTION

We consider the problem of sampling from a log-concave distribution supported on a polytope: Given a polytope $K := \{\theta \in \mathbb{R}^d : A\theta \leq b\}$, where $A \in \mathbb{R}^{m \times d}$ and $b \in \mathbb{R}^m$, and a convex function $f : K \to \mathbb{R}$, output a sample $\theta \in K$ from the distribution $\pi(\theta) \propto e^{-f(\theta)}$. This problem arises in many applications, including in Bayesian inference, differentially private optimization, and integration. For instance, in Bayesian Lasso logistic regression, $f(\theta) = \sum_{i=1}^{n} \ell(\theta^\top x_i)$ where $\ell$ is the logistic loss and $x_i$ are datapoints with $\|x_i\|_2 \leq 1$, and $K = \{\theta \in \mathbb{R}^d : \|\theta\|_1 \leq O(1)\}$ is an $\ell_1$-ball (see e.g., Tian et al. (2008); Zhang et al. (2017); Kim et al. (2018)). The closely related optimization problem of minimizing a $L$-Lipschitz or $\beta$-smooth convex function $f$ constrained to a polytope arises in numerous contexts as well. A special case is the setting where $f$ is linear on $K$, which corresponds to linear programming, since in this case $\beta = 0$. In applications where one must preserve privacy when minimizing $f$ constrained to $K$, sampling from the exponential mechanism of McSherry & Talwar (2007), where $\pi(\theta) \propto e^{-\frac{1}{\varepsilon}f(\theta)}$ on $K$, allows one to obtain optimal utility bounds under $\varepsilon$-differential privacy (see also Bassily et al. (2014); Munoz et al. (2021); Kapralov & Talwar (2013)). Depending on the application, the polytope $K$ could be e.g. the probability simplex, hypercube, or $\ell_1$-ball, where $m = O(d)$ and $\mathrm{nnz}(A) = O(d)$, as well as many applications where the number of constraints defining the polytope is $m \geq d^2$ and $\mathrm{nnz}(A) = O(m)$ (see e.g. Hsu et al. (2014), Barvinok & Rudelson (2021) for some examples). Here $\mathrm{nnz}(M)$ denotes the number of non-zero entries of a matrix $M$.

This sampling problem, and its generalization to convex $K$, has been studied in multiple prior works. One line of work Kannan & Narayanan (2012) Narayanan (2016), Lee & Vempala (2018), Cousins & Vempala (2018), Chen et al. (2017), Jia et al. (2021) focuses on the special case ($f \equiv 0$). In particular, Narayanan (2016) gives an algorithm, the Dikin walk Markov chain, which takes $O(md \times md^{\omega-1}) \times \log(\frac{w}{\delta})$ arithmetic operations to sample within total variation (TV) error $\delta > 0$ from the uniform distribution on a polytope $K \subseteq \mathbb{R}^d$ given by $m$ inequalities from a $w$-warm start. Here, $\omega = 2.37 \cdots$ is the matrix-multiplication constant and a distribution $\mu$ is $w$-warm for $w \geq 1$ w.r.t.

the target distribution $\pi$ if $\mu(z)/\pi(z) \le w$ for every $z \in K$. Other works consider the more general problem when $\pi$ may not be the uniform distribution. Multiple works Applegate & Kannan (1991); Frieze et al. (1994); Frieze & Kannan (1999); Lovász & Vempala (2006; 2007); Brosse et al. (2017) have given algorithms that apply in the general setting when $K$ is a convex body with access given by a membership oracle (or related oracles).

Narayanan & Rakhlin (2017) give an algorithm based on the Dikin-walk Markov chain to sample from any log-concave $\pi \propto e^{-f}$ on $K$ where $f$ is $L$-Lipschitz or $\beta$-smooth. Building upon the work of Narayanan & Rakhlin (2017), Mangoubi & Vishnoi (2022) presented a Dikin walk which utilizes a "soft-threshold" regularizer, which takes $O((md + dL^2R^2) \times \log(\frac{w}{\delta}) \times (md^{\omega-1} + T_f))$ arithmetic operations to sample from an $L$-log-Lipschitz log-concave distribution $\pi$ on a polytope $K$ within error $\delta > 0$ in the TV distance from a $w$-warm start. Here $T_f$ is the number of arithmetic operations to compute the value of $f$. Specifically, each iteration in their soft-threshold Dikin walk algorithm requires $md^{\omega-1}$ arithmetic operations to compute the (inverse) Hessian of the log-barrier function for the polytope $K$ and its determinant.

**Our contributions.** We show that the per-iteration cost of computing the (inverse) Hessian of the log-barrier with soft-threshold regularizer, and its determinant, at each step of the soft-threshold Dikin walk can be reduced from $md^{\omega-1}$ to $\mathrm{nnz}(A) + d^2$ (see Theorem 2.1). More specifically, our version of the Dikin walk algorithm (Algorithm 1) takes $\tilde{O}((md + dL^2R^2) \log(\frac{w}{\delta})) \times (\mathrm{nnz}(A) + d^2 + T_f)$ arithmetic operations to sample within total variation error $O(\delta)$ from an $L$-log-Lipschitz log-concave distribution on a polytope $K$ defined by $m$ inequalities, and $\tilde{O}((md + d\beta R^2) \log(\frac{w}{\delta})) \times (\mathrm{nnz}(A) + d^2 + T_f)$ arithmetic operations to sample from a $\beta$-log-smooth log-concave distribution on $K$. Compared to the implementation of the soft-threshold Dikin walk in Mangoubi & Vishnoi (2022), we obtain an improvement in the runtime for the case of $L$-log-Lipschitz or $\beta$-log-smooth log-concave distributions of at least $d^{\omega-2}$ in all cases where, e.g., $T_f = \Omega(d^2)$. If one also has $\mathrm{nnz}(A) = O(m)$, the improvement is $d^{\omega-1}$ if $m \ge d^2$ and $md^{\omega-3}$ if $m \le d^2$; see Table 1 for comparison to prior works.

The main challenge in improving the per-step complexity of the soft-threshold Dikin walk is that the current algorithm uses dense matrix multiplication to compute the Hessian of the log-barrier function with soft-threshold regularization, as well as its determinant, which requires $md^{\omega-1}$ arithmetic operations. The fact that the Hessian of the log-barrier function oftentimes changes slowly at each step of interior point-based methods was used by Karmarkar (1984), Vaidya (1989), and Nesterov & Nemirovsky (1991), and later by Lee & Sidford (2015), to develop faster linear solvers for the Hessian to improve the per-iteration complexity of interior point methods for linear programming. To obtain similar improvements in the computation time of the soft-threshold Dikin walk, we need to show that the regularized barrier function proposed by Mangoubi & Vishnoi (2022) changes slowly as well. The notion of a barrier function changing slowly with respect to the Frobenius norm proposed in Laddha et al. (2020) suffices, but we have to show that this notion holds for the soft-threshold regularized log-barrier function. To improve the per-iteration complexity of the soft-threshold Dikin walk, we show how to use the efficient inverse maintenance algorithm of Lee & Sidford (2015) to maintain a linear system solver for the Hessian of the log-barrier function with soft-threshold regularizer, and then show how to use this solver to efficiently compute a random estimate whose expectation is the Hessian determinant in $\mathrm{nnz}(A) + d^2$ arithmetic operations. This is accomplished by first computing a randomized estimate for the log-determinant of the Hessian, and then converting this estimate into a randomized estimate for a "smoothed" Metropolis acceptance probability via a piecewise-polynomial series expansion for the acceptance probability. We present a detailed overview of the techniques in Section 4.

## 2 MAIN RESULT

**Theorem 2.1 (Main result)** *There exists an algorithm (Algorithm 1) which, given the following inputs, 1) $\delta, R > 0$ and either $L > 0$ or $\beta > 0$, 2) $A \in \mathbb{R}^{m \times d}$, $b \in \mathbb{R}^m$ that define a polytope $K := \{\theta \in \mathbb{R}^d : A\theta \le b\}$ such that $K$ is contained in a ball of radius $R$ and has nonempty interior, 3) an oracle for the value of a convex function $f : K \to \mathbb{R}$, where $f$ is either $L$-Lipschitz or $\beta$-smooth, and 4) an initial point sampled from a distribution supported on $K$ which is $w$-warm with respect to $\pi \propto e^{-f}$ for some $w > 0$, outputs a point from a distribution $\mu$ where $\|\mu - \pi\|_{\mathrm{TV}} \le \delta$. This algorithm takes at most*

- $\tilde{O}((md+dL^2R^2)\log(\frac{w}{\delta}))\times(\mathrm{nnz}(A)+d^2+T_f)$ *arithmetic operations in the setting where* $f$ *is L-Lipschitz,*

- *or* $\tilde{O}((md+d\beta R^2)\log(\frac{w}{\delta}))\times(\mathrm{nnz}(A)+d^2+T_f)$ *arithmetic operations in the setting where* $f$ *is* $\beta$-*smooth,*

*where* $T_f$ *is the number of arithmetic operations to evaluate* $f$.

Theorem 2.1 improves by a factor of $\frac{md^{\omega-1}+T_f}{\max(\mathrm{nnz}(A),d^2)+T_f}$ arithmetic operations on the previous bound of Mangoubi & Vishnoi (2022) of $\tilde{O}((md+dL^2R^2)\times(md^{\omega-1}+T_f))$ arithmetic operations for sampling from a distribution $\propto e^{-f}$ where $f$ is $L$-Lipschitz from a $w$-warm start. Here $\tilde{O}$ hides logarithmic factors in $\delta, w, L, R, d$. When $f$ is instead $\beta$-smooth their bound is $\tilde{O}((md+d\beta R^2)\times(md^{\omega-1}+T_f))$, and the improvement is the same. When $T_f \le O(d^2)$ (as may be the case when evaluating $f$ requires computing inner products with $n=d$ datapoints in $\mathbb{R}^d$), the improvement is (at least) $d^{\omega-2}$. If one also has that $A$ is $O(m)$-sparse, $(\mathrm{nnz}(A)=O(m))$, as is the case in applications where each constraint inequality involves $O(1)$ variables), the improvement is $d^{\omega-1}$ in the regime $m \ge \Omega(d^2)$ and $md^{\omega-3}$ if $m \le O(d^2)$.

Theorem 2.1 also improves on the bound of (Lovász & Vempala (2006); Theorem 1.1) of $\tilde{O}(d^2(R/r)^2)(md+T_f)$ arithmetic operations by $\min((\frac{R}{r})^2, \frac{d}{r^2L^2})$ when e.g. $T_f = \Omega(d^2)$ and $m = O(d)$. In the setting where $f$ is $\beta$-smooth, the improvement is $\min((\frac{R}{r})^2, \frac{d}{r^2\beta})$.

In the Bayesian Lasso logistic regression example considered in Mangoubi & Vishnoi (2022), our algorithm takes $\tilde{O}(d^4)$ operations from a $w$-warm start as $f$ is $\beta$-smooth and $L$-Lipschitz, with $\beta = L = n = m = d$, $R = 1$, $r = 1/\sqrt{d}$, $\mathrm{nnz}(A) = O(d)$. This improves by $d^{\omega-2}$ on their bound of $\tilde{O}(d^{2+\omega})$, and by $d$ on the bound of $\tilde{O}(d^5)$ implied by Lovász & Vempala (2006).

Theorem 2.1 also improves upon the running time for $(\varepsilon, 0)$-differentially private low-rank approximation by a factor of $d^{\omega-2}$ on the bound of $O(d^{2+\omega}\log(\frac{w}{\delta}))$ operations of Mangoubi & Vishnoi (2022). Moreover, it improves by a factor of $\frac{md^{\omega-1}}{\mathrm{nnz}(A)+d^2}$ on their bound of $O((md+dn^2\varepsilon^2)(\varepsilon n + d\log(nRd/(r\varepsilon))(md^{\omega-1}+T_f))$ arithmetic operations for the problem of finding a minimizer $\hat{\theta} \in K$ of a convex empirical risk function $f(\theta, x) = \sum_{i=1}^n \ell_i(\theta, x_i)$ under $(\varepsilon, 0)$-differential privacy, when the losses $\ell_i(\cdot, x)$ are $\hat{L}$-Lipschitz, for any $\hat{L} > 0$ and dataset $x \in \mathcal{D}^n$. See the details in their paper.

| Algorithm | Arithmetic operations per iteration | Iterations for $L$-Lipschitz $f$ | Iterations for $\beta$-smooth $f$ | Arithmetic operations if $m, \mathrm{nnz}(A) = O(d^2)$, $L, r = O(1), R = O(d)$, $T_f = O(d^2)$ | Arithmetic operations if $m = O(d), r = O(\frac{1}{\sqrt{d}})$ $\beta = L = O(d)$, $R = O(1), T_f = O(d^2)$ |
|---|---|---|---|---|---|
| Proximal Langevin MC (Brosse et al. (2017)) | $O(md^{\omega-1})+T_{\nabla f}$ | $\tilde{O}(d^5\delta^{-6}L^2(\frac{R}{r})^4)$ | — | $\tilde{O}(d^{10+\omega}\delta^{-6})$ | $\tilde{O}(d^{9+\omega}\delta^{-6})$ |
| Hit-and-run (Lovász & Vempala (2006)) | $O(md)+T_f$ | $\tilde{O}(d^2(\frac{R}{r})^2)$ | same | $\tilde{O}(d^7)$ | $\tilde{O}(d^5)$ |
| Dikin Walk of Narayanan & Rakhlin (2017) | $O(md^{\omega-1})+T_f$ | $\tilde{O}(d^5+d^3L^2R^2)$ | $\tilde{O}(d^5+d^3\beta R^2)$ | $\tilde{O}(d^{6+\omega})$ | $\tilde{O}(d^{5+\omega})$ |
| Soft-Threshold Dikin Walk (Mangoubi & Vishnoi (2022)) | $O(md^{\omega-1})+T_f$ | $\tilde{O}(md+dL^2R^2)$ | $\tilde{O}(md+d\beta R^2)$ | $\tilde{O}(d^{5+\omega})$ | $\tilde{O}(d^{2+\omega})$ |
| **This paper** | $O(\mathrm{nnz}(A)+d^2)+T_f$ | $\tilde{O}(md+dL^2R^2)$ | $\tilde{O}(md+d\beta R^2)$ | $\tilde{O}(d^6)$ | $\tilde{O}(d^4)$ |

Table 1: Number of iterations (and arithmetic operations per-iteration) of different algorithms which imply bounds for sampling within TV error $O(\delta)$ from a logconcave $\pi \propto e^{-f}$ on a polytope $K$ when $f$ is L-Lipschtiz or $\beta$-smooth, from a $w$-warm start. $T_f$ and $T_{\nabla f}$ are, respectively, the number of operations to compute $f$ or $\nabla f$. The $\tilde{O}$ notation hides logarithmic factors of $d, \delta, r, R, w$. The fifth column gives runtimes when $K$ is a polytope with $m, \mathrm{nnz}(A) = O(d^2)$ and $R = O(d)$ that contains the unit ball (and is thus not well-rounded), and $f$ is $O(1)$-Lipschitz. The sixth column corresponds to sampling a Bayesian Lasso logistic regression posterior distribution with $O(d)$ datapoints, where $K$ is the unit $\ell_1$-ball.

# 3 ALGORITHM

For input $\theta \in \mathrm{Int}(K)$, let $S(\theta) := \mathrm{Diag}(A\theta - b)$, and $\Phi(\theta) := \alpha^{-1}\hat{A}^\top\hat{S}(\theta)^2\hat{A}$, where $\hat{S}(\theta) := \begin{pmatrix} S(\theta) & 0 \\ 0 & \alpha^{\frac{1}{2}}\eta^{-\frac{1}{2}}I_d \end{pmatrix}$ and $\hat{A} := \begin{pmatrix} A \\ I_d\cdot \end{pmatrix}$ For input $W \in \mathbb{R}^{d\times d}$ and $t \in [0,1]$, let $\Gamma(W, t) := I_{m+d} + t(W - I_{m+d})$. For a rectangular matrix $M$, a vector $v \in \mathbb{R}^d$, a diagonal matrix $D$, and a set of algorithmic parameters $P$, **Solve**$(v, D, M; P)$ outputs a vector $w$ which is (ideally)

a solution to the system of linear equations $M^\top D M w = v$. Given an initial diagonal matrix $D_0$, **Initialize**$(D_0, M)$ generates parameters $P$ corresponding to $D_0$. Given a new diagonal matrix $D$, and a set of parameters $P$, **Update**$(D, M; P)$ updates the parameters $P$ and outputs some new set of parameters $P'$ corresponding to the new diagonal matrix $D$. For any $S \subset \mathbb{R}^d$ denote the interior of $S$ by $\mathrm{Int}(S) := \{\theta \in S : B(\theta, t) \subseteq S \text{ for some } t > 0\}$.

---

**Algorithm 1:** Fast implementation of soft-threshold Dikin walk

---

**Input:** $m, d \in \mathbb{N}$, $A \in \mathbb{R}^{m \times d}$, $b \in \mathbb{R}^m$, which define the polytope $K := \{\theta \in \mathbb{R}^d : A\theta \le b\}$
**Input:** Oracle returning the value of a convex function $f : K \to \mathbb{R}$. Initial point $\theta_0 \in \mathrm{Int}(K)$

1 **Hyperparameters:** $\alpha > 0; \eta > 0; T \in \mathbb{N}; \mathcal{N} \in \mathbb{N}; \gamma > 0$

2 Set $\theta \leftarrow \theta_0$, $\hat{A} = \begin{pmatrix} A \\ I_d, \end{pmatrix}$, $a = \log\det(\hat{A}^\top \hat{A})$, $P = $ **Initialize**$(\hat{S}(\theta)^2, \hat{A})$, $Q_0 = $ **Initialize**$(I_d, \hat{A})$

3 **for** $i = 1, \dots, T$ **do**

4     Sample a point $\xi \sim N(0, I_d)$

5     Set $u = \hat{A}^\top \hat{S}(\theta)\xi$

6     Set $z = \alpha^{-\frac{1}{2}}$ **Solve**$(u, \hat{S}(\theta)^2, \hat{A}; P)$ {*Computes Gaussian $z$ with covariance $\alpha^{-1}\hat{A}^\top \hat{S}(\theta)^2 \hat{A}$*}

7     **if** $z \in \mathrm{Int}(K)$ **then**

8        Set $W = \hat{S}(\theta)^{-2}\hat{S}(z)$

9        **for** $j = 1, \dots, \mathcal{N}$ **do**

10           Sample $v \sim N(0, I_d)$

11           Sample $t$ from the uniform distribution on $[0, 1]$.

12           Set $Q = $ **Update**$(\Gamma(W, t), \hat{A}; Q_0)$

13           Set $w = \hat{A}^\top (W - I_d)\hat{A}v$

14           Set $Y_j = v^\top$ **Solve**$(w, \Gamma(W, t), \hat{A}; Q) + a$

15           {*Computes a random variable $Y_j$ with expectation $\log\det\Phi(z) - \log\det\Phi(\theta)$*}

16        **end**

17        **if** $\frac{1}{4} \le Y_1 < 2\log\frac{1}{\gamma}$ **then**

18           Set $X = 1 + \frac{1}{2}\sum_{k=1}^{2\mathcal{N}-1}\sum_{\ell=0}^{\mathcal{N}}\left((-1)^k \frac{1}{\ell!}\prod_{j=1}^{\ell}(-2kY_j)\right)$

19        **if** $Y_1 < \frac{1}{4}$ **then**

20           Set $c_0, \dots, c_{\mathcal{N}}$ to be the first $\mathcal{N}$ coefficients of the Taylor expansion of $\frac{1}{1+e^{-t}}$ at $0$

21           Set $X = \sum_{\ell=0}^{\mathcal{N}}\left(c_\ell \prod_{j=1}^{\ell}(Y_j)\right)$

22        **if** $Y_1 \ge 2\log(\frac{1}{\gamma})$ **then**

23           Set $X = 1$

24        **end**

25        Accept $\theta \leftarrow z$ with probability $\frac{1}{2}\min(\max(X, 0), 1)\min\left(\frac{e^{-f(z)}}{e^{-f(\theta)}}e^{\|z-\theta\|_{\Phi(\theta)}^2 - \|\theta-z\|_{\Phi(z)}^2}, 1\right)$

26        Set $P = $ **Update**$(\hat{S}(\theta)^2, \hat{A}; P)$

27     **else**

28        Reject $z$

29     **end**

30 **end**

31 **Output:** $\theta$

---

In Theorem 1, we set $\gamma = \delta \frac{1}{10^{20}(md+L^2R^2)}\log^{1.02}(w/\delta)$. We set the step size hyperparameters $\alpha = 1/(10^5 d)\log^{-1}(1/\gamma)$ and $\eta = 1/(10^4 dL^2)$ if $f$ is $L$-Lipschitz, and the number of steps to be $T = 10^9 \left(2m\alpha^{-1} + \eta^{-1}R^2\right) \times \log(w/\delta) \times \log^{1.01}(10^9 \left(2m\alpha^{-1} + \eta^{-1}R^2\right) \times \log(w/\delta))$ When $f$ is $\beta$-smooth (but not necessarily Lipschitz), we instead set $\gamma = \delta \frac{1}{10^{20}(md+\beta R^2)}\log^{1.01}(w/\delta)$, and $\alpha = 1/(10^5 d)\log^{-1}(1/\gamma)$ and $\eta = 1/(10^4 d\beta)$. Thus, in either case, we have $\gamma \le \delta/(1000T)$. Finally, we set the parameter $\mathcal{N}$, the number of terms in the Taylor expansions, to be $\mathcal{N} = 10\log(\frac{1}{\gamma})$.

## 4 OVERVIEW OF PROOF OF MAIN RESULT

We give an overview of the proof of Theorem 2.1. Suppose we are given $A \in \mathbb{R}^{m \times d}$, $b \in \mathbb{R}^m$ that define a polytope $K := \{\theta \in \mathbb{R}^d : A\theta \le b\}$ with non-empty interior and contained in a ball of radius $R > 0$, and an oracle for the value of a convex function $f : K \to \mathbb{R}$, where $f$ is either

$L$-Lipschitz or $\beta$-smooth. Our goal is to sample from $\pi \propto e^{-f}$ on $K$ within any TV error $\delta > 0$ in a number of arithmetic operations which improves on the best previous runtime for this problem (when, e.g., $L, \beta, R = O(1)$), which was given for the soft-threshold Dikin walk Markov chain. We do this by showing how to design an algorithm to implement the soft-threshold Dikin walk Markov chain with faster per-iteration complexity.

**Dikin walk in the special case** $f \equiv 0$. First, consider the setting where one wishes to sample from the uniform distribution ($f \equiv 0$) on a polytope $K$, where $K$ is contained in a ball of radius $R > 0$ and contains a ball of smaller radius $r > 0$.

Kannan & Narayanan (2012) give a Markov chain-based algorithm, the Dikin walk, which uses the log-barrier function from interior point methods to take large steps while still remaining inside the polytope $K$. The log-barrier $\varphi(\theta) = -\sum_{j=1}^{m} \log(b_j - a_j^\top \theta)$ for the polytope $K$ defines a Riemannian metric with associated norm $\|\theta\|_{H(\theta)} := \sqrt{u^\top H(\theta) u}$, where $H(\theta) := \nabla^2 \varphi(\theta) = \sum_{j=1}^{m} \frac{a_j a_j^\top}{(b_j - a_j^\top \theta)^2}$ is the Hessian of the log-barrier function. At any point $\theta \in \text{Int}(K)$, the unit ball with respect to this norm, $E(\theta) := \{z \in \mathbb{R}^d : \|z - \theta\|_{H(\theta)} \leq 1\}$, referred to as the Dikin ellipsoid, is contained in $K$. To sample from $K$, from any point $\theta \in \text{Int}(K)$, the Dikin walk proposes steps $z$ from a Gaussian with covariance $\alpha H^{-1}(\theta)$, where $\alpha > 0$ is a hyperparameter. If one chooses $\alpha \leq O(\frac{1}{d})$, this Gaussian concentrates inside $E(\theta) \subseteq K$, allowing the Markov chain to propose steps which remain inside $K$ w.h.p.

To ensure the stationary distribution of the Dikin walk is the uniform distribution on $K$, if $z \in \text{Int}(K)$, the Markov chain accepts each proposed step $z$ with probability determined by the Metropolis acceptance rule $\min(\frac{p(z \to \theta)}{p(\theta \to z)}, 1) = \frac{\sqrt{\det(H(\theta))}}{\sqrt{\det(H(z))}} e^{\|\theta - z\|_{H(\theta)}^2 - \|z - \theta\|_{H(z)}^2}$, where $p(\theta \to z) \propto (\det H(\theta))^{-\frac{1}{2}} e^{-\|\theta - z\|_{H(\theta)}^2}$ denotes the probability (density) that the Markov chain at $\theta$ proposes an update to the point $z$.

If one chooses $\alpha$ too large, each step of the Markov chain will be rejected with high probability. Thus, ideally, one would like to choose $\alpha$ as large as possible such that the proposed steps are accepted with probability $\Omega(1)$. To bound the terms $\sqrt{\det(H(z))}/\sqrt{\det(H(\theta))}$ and $e^{\|z - \theta\|_{H(z)}^2 - \|\theta - z\|_{H(\theta)}^2}$, Kannan & Narayanan (2012) use the fact that the Hessian of the log-barrier changes slowly w.r.t. the local norm $\|\cdot\|_{H(\theta)}$. More specifically, to bound the change in the determinantal term, they use the following property of the log-barrier, discovered by Vaidya & Atkinson (1993), which says its log-determinant $V(\theta) := \log \det H(\theta)$ satisfies

$$(\nabla V(\theta))^\top [H(\theta)]^{-1} \nabla V(\theta) \leq O(d) \qquad \forall \theta \in \text{Int}(K).$$

Since the proposed update $(z - \theta)$ is Gaussian with covariance $\alpha H^{-1}(\theta)$, if one chooses $\alpha = \frac{1}{d}$, by standard Gaussian concentration inequalities $(\theta - z)^\top \nabla V(\theta) \leq O(1)$ w.h.p. Thus, $\log \det(H(z)) - \log \det(H(\theta)) = V(z) - V(\theta) = \Omega(1)$, and $\sqrt{\det(H(z))}/\sqrt{\det(H(\theta))} = \Omega(1)$ w.h.p.

To bound the total variation distance between the target (uniform) distribution $\pi$ and the distribution $\nu_T$ of the Markov chain after $T$ steps, Kannan & Narayanan (2012) use an isoperimetric inequality of Lovász & Vempala (2003) which is defined in terms of the Hilbert distance on the polytope $K$. For any distinct points $u, v \in \text{Int}(K)$ the Hilbert distance is

$$\sigma(u, v) := \frac{\|u - v\|_2 \times \|p - q\|_2}{\|p - u\|_2 \times \|v - q\|_2},$$

where $p, q$ are endpoints of the chord through $K$ passing through $u, v$ in the order $p, u, v, q$.

To apply this isoperimetric inequality to the Dikin walk, which proposes steps of size roughly $O(\alpha)$ w.r.t. the local norm $\|\cdot\|_{\alpha^{-1} H(\theta)}$, they show that the Hilbert distance satisfies

$$\sigma^2(u, v) \geq \frac{1}{m} \sum_{i=1}^{m} \frac{(a_i^\top (u - v))^2}{(a_i^\top u - b_i)^2} \geq \frac{1}{m\alpha^{-1}} \|u - v\|_{\alpha^{-1} H(\theta)}^2. \tag{1}$$

They then use the isoperimetric inequality w.r.t. the Hilbert distance together with standard conductance arguments for bounding the mixing time of Markov chains (see e.g. Lovász & Simonovits (1993)), to show that, if the Markov chain is initialized with a $w$-warm start, after $i$ steps, the total variation distance to the stationary distribution $\pi$ satisfies

$$\|\nu_i - \pi\|_{\text{TV}} \leq O(\sqrt{w}(1 - m\alpha^{-1})^i) \qquad \forall i \in \mathbb{N},$$

provided that each step proposed by the Markov chain is accepted with probability $\Omega(1)$. For $\alpha = O(\frac{1}{d})$, after $T = O(m\alpha^{-1}\log(\frac{w}{\delta})) = O(md\log(\frac{w}{\delta}))$ steps, $\|\nu_T - \pi\|_{\text{TV}} \le \delta$.

At each step in their algorithm they compute the proposed update $z = \alpha^{-\frac{1}{2}}\sqrt{H^{-1}(\theta)}\xi$ where $\xi \sim N(0, I_d)$, and the terms $\sqrt{\det(H(z))}/\sqrt{\det(H(\theta))}$ and $e^{\|z-\theta\|^2_{H(z)} - \|\theta-z\|^2_{H(\theta)}}$ in the acceptance probability. Writing $H(\theta) = A^\top S(\theta)^{-2}A$ where $S(\theta) = \text{Diag}(A\theta - b)$, these computations can be done in $O(md^{\omega-1})$ arithmetic operations using dense matrix multiplication. Their algorithm's runtime is then $T \times md^{\omega-1} = O(md \times md^{\omega-1} \times \log(\frac{w}{\delta}))$ arithmetic operations.

**Dikin walks for sampling from Lipschitz/smooth log-concave distributions.** One can extend the Dikin walk to sample from more general log-concave functions on a polytope $K$, by introducing a term $e^{f(\theta)-f(z)}$ to the Metropolis acceptance probability. This causes the Markov chain to have stationary distribution $\pi(\theta) \propto e^{-f(\theta)}$, since for every $\theta, z \in \text{Int}(K)$,

$$\min\left(\tfrac{p(z\to\theta)}{p(\theta\to z)}e^{f(\theta)-f(z)}, \; 1\right)p(\theta \to z)\pi(\theta) = \min\left(\tfrac{p(\theta\to z)}{p(z\to\theta)}e^{f(z)-f(\theta)}, \; 1\right)p(z \to \theta)\pi(z).$$

However, the term $e^{f(\theta)-f(z)}$ can become exponentially small with the distance $\|z - \theta\|_2$ (and thus with the step-size $\alpha$) even if, e.g., $f$ is $L$-Lipschitz. Thus, to obtain a runtime polynomial in $L, d$, one may need to re-scale the covariance $\alpha H^{-1}(\theta)$ to ensure $e^{f(\theta)-f(z)} \ge \Omega(1)$.

One approach, taken by Mangoubi & Vishnoi (2022), is to propose steps with (inverse) covariance matrix equal to a regularized log-barrier Hessian, $\Phi(\theta) = \alpha^{-1}H(\theta) + \eta^{-1}I_d$, for some $\eta > 0$. Setting $\eta = \frac{1}{dL^2}$ ensures (by standard Gaussian concentration inequalities) that the proposed step $z$ satisfies $\|z - \theta\|_2 \le O(1)$ w.h.p. and hence $e^{f(\theta)-f(z)} = \Omega(1)$ w.h.p.

To bound the total variation distance, in place of (1), they show a lower bound on the Hilbert distance w.r.t. the local norm $\|\cdot\|_{\Phi(u)}$,

$$\sigma^2(u,v) \ge \tfrac{1}{2}\tfrac{\|u-v\|_2^2}{R^2} + \tfrac{1}{2m}\sum_{i=1}^m \tfrac{(a_i^\top(u-v))^2}{(a_i^\top u - b_i)^2} \ge \tfrac{1}{2m\alpha^{-1} + 2\eta^{-1}R^2}\|u-v\|^2_{\Phi(u)}.$$

This leads to a bound of $O(m\alpha^{-1} + \eta^{-1}R^2)\log(\frac{w}{\delta})) = O(md + L^2R^2)\log(\frac{w}{\delta}))$ on the number of steps until the Markov chain is within $O(\delta)$ of $\pi$ in the total variation distance. In addition to the operations required in the special case when $f \equiv 0$, each step requires calling the oracle for $f$ which takes $T_f$ arithmetic operations. Thus, they obtain a runtime of $O((md + L^2R^2) \times (md^{\omega-1} + T_f) \times \log(\frac{w}{\delta}))$ arithmetic operations to sample from $\pi \propto e^{-f}$.

**Faster implementation of matrix operations when $f \equiv 0$.** When $f \equiv 0$, one can use that the log-barrier Hessian $H(\theta)$ changes slowly at each step of the Dikin walk to compute the proposed step, and determinantal terms in the acceptance probability, more efficiently.

The fact that the Hessian of the log-barrier function oftentimes changes slowly at each step of interior point-based methods was used by Karmarkar (1984), and later by Vaidya (1989), and Nesterov & Nemirovsky (1991) when developing faster linear solvers for $H(\theta)$ to use in interior point methods for linear programming. Specifically, they noticed for many interior point methods, which take steps $\theta_1, \theta_2, \ldots \in \text{Int}(K)$, the update to $H(\theta_i) = A^\top D_i A$ at each iteration $i$, where $D_i := S(\theta_i)$, is (nearly) low-rank, since in these interior point methods most of the entries of the diagonal matrix $D_{i+1}^{-1}D_i$ are very small at each step $i$. This allowed them to use the Woodbury matrix formula to compute a low-rank update for the inverse Hessian $H(\theta_i)$, reducing the per-step complexity of maintaining a linear solver for $H(\theta_i)$ at each iteration $i$. Specifically, the Woodbury matrix formula says

$$(M + UCV)^{-1} = M^{-1} - M^{-1}U(C^{-1} + VM^{-1}U)^{-1}V^{-1}M), \tag{2}$$

for any $M \in \mathbb{R}^{d\times d}, U \in \mathbb{R}^{d\times k}, V \in \mathbb{R}^{k\times d}, C \in \mathbb{R}^{k\times k}$, where $k$ may be much smaller than $d$.

Lee & Sidford (2015) obtain an improved algorithm for maintaining linear solvers for sequences of matrices $\{A^\top D_i A\}_{i=1,2,\ldots}$ under the assumption that $D_i$ changes slowly at each iteration $i$ with respect a weighted Frobenius norm, $\|(D_{i+1} - D_i)D_i^{-1}\|_F \le O(1)$. This assumption is weaker than requiring most of the entries to be nearly constant at each step, as it only requires a (weighted) sum-of-squares of the change in the entries of $D_i$ to be $O(1)$. To obtain their method, they sample diagonal entries of $D_i$ with probability proportional to the (change in) the entries' leverage scores, a quantity used to measure the importance of the rows of a matrix (for a matrix $M$ the $i$'th leverage

score is defined as $[M(M^\top M)^{-1}M^\top]_{ii}$). Using these subsampled diagonal entries they show how to obtain a spectral sparsifier for the updates $A^\top(D_{i+1} - D_i)A$, allowing them to make low-rank updates to the matrix $A^\top D_i A$ at each iteration. Specifically, for any sequence of diagonal matrices $D_1, D_2, \ldots$ satisfying

$$\|(D_{i+1} - D_i)D_i^{-1}\|_F \leq O(1) \tag{3}$$

at each $i \in \mathbb{N}$, they show that they can maintain a $O(\mathrm{nnz}(A) + d^2)$-time linear system solver for the matrices $A^\top D_i A$, at a cost of $O(\mathrm{nnz}(A) + d^2)$ arithmetic operations per-iteration. (plus an initial cost of $O(\mathrm{nnz}(A) + d^\omega)$ to initialize their algorithm).

Laddha et al. (2020) apply the fast linear solver of Lee & Sidford (2015) to the Dikin walk in the special case where $f \equiv 0$. Specifically, they show the sequence of log-barrier Hessians $A^\top D_i A$ where $D_i = S^2(\theta_i)$, also satisfies (3) when $\theta_i$ are the steps of the Dikin walk. They then use the fast linear solver of Lee & Sidford (2015) to evaluate the matrix operations $(A^\top S(\theta_i)^{-2}A)^{-\frac{1}{2}}\xi$ and compute an estimate for $\det(A^\top S(\theta_i)^{-2}A)$ at each iteration. This gives a per-iteration complexity of $\mathrm{nnz}(A) + d^2$ for the Dikin walk in the special case $f \equiv 0$.

**Our work.** To obtain a faster implementation of the (soft-threshold) Dikin walk in the general setting where $f$ is Lipschitz or smooth on $K$, we would ideally like to apply the efficient inverse maintenance algorithm of Lee & Sidford (2015) to obtain a fast linear solver for the log-barrier Hessian with soft-threshold regularizer $\Phi(\theta) = \alpha^{-1}A^\top S(\theta)^{-2}A + \eta I_d$. Specifically, writing $\Phi(\theta) = \alpha^{-1}\hat{A}^\top \hat{S}(\theta)^2 \hat{A}$ where $\hat{A} := \begin{pmatrix} A \\ I_{d,} \end{pmatrix}$ and $\hat{S}(\theta) := \begin{pmatrix} S(\theta) & 0 \\ 0 & \alpha^{\frac{1}{2}}\eta^{-\frac{1}{2}}I_d \end{pmatrix}$, we would like to obtain a fast linear solver for the sequence of matrices $\Phi(\theta_1), \Phi(\theta_2), \ldots$, where $\theta_1, \theta_2, \cdots$ are the steps of the soft-threshold Dikin walk, and use it to improve the per-iteration complexity of the walk. This poses two challenges:

1. To obtain an $O(\mathrm{nnz}(A) + d^2)$-time linear solver with the efficient inverse maintenance algorithm, we need to show that $\Phi(\theta_i) = \alpha^{-1}\hat{A}^\top \hat{S}(\theta_i)^2 \hat{A}$ does not change too quickly at each step $i$ of the soft-threshold Dikin walk. Specifically, we need to show, at each $i$ w.h.p.,

$$\|(\hat{S}^2(\theta_{i+1}) - \hat{S}^2(\theta_{i+1}))\hat{S}^{-2}(\theta_{i+1})\|_F \leq O(1). \tag{4}$$

Once we have a linear solver for $\Phi(\theta)$, this immediately gives us a way to compute the proposed updates of the Markov chain by solving a system of linear equations (see e.g. Section 2.1.1 of Kannan & Narayanan (2012)).

2. We also need to evaluate the terms $\det(\Phi(\theta))$ in the acceptance probability. However, access to a linear solver for $\Phi(\theta)$ does not directly give a way to compute its determinant.

**Bounding the change in the soft-threshold log-barrier Hessian.** From concentration inequalities for multivariate Gaussian distributions, one can show that, with high probability, the Dikin walk proposes updates $z \leftarrow \theta$ which have length $O(\sqrt{d})$ in the local norm: $\|z - \theta\|_{\Phi(\theta)} = O(\sqrt{d})$ w.h.p. To bound the change in the soft-threshold-regularized log-barrier Hessian matrix $\Phi(\theta)$ at each step of the walk, we would ideally like to show that the Frobenius norm of the derivative of this matrix changes slowly with respect to the local norm $\|\cdot\|_{\Phi(\theta)}$. The soft-threshold-regularized log-barrier function $\Phi$ satisfies the self-concordance property $D\Phi(\theta)[h] \preceq 2\sqrt{v^\top \Phi(\theta)v}\Phi(\theta)$ for any $\theta, z \in \mathrm{Int}(K)$, and $v \in \mathbb{R}^d$, where $DM(\theta)[h]$ denotes the derivative of a matrix-valued function $M(\theta)$ in the direction $h$. However, the self-concordance property does not directly imply that (4) holds, as (4) contains a bound in the Frobenius norm. To overcome this issue, we show that (a rescaling of) the regularized log-barrier Hessian, $\Psi(\theta) := \alpha^{\frac{1}{2}}\Phi(\theta) = \nabla^2\varphi(\theta) + \alpha^{\frac{1}{2}}\eta^{-\frac{1}{2}}I_d$ satisfies the following strengthening of the self-concordance property w.r.t. the Frobenius norm:

$$\|(\Psi(z) - \Psi(\theta))^{-1}\Psi^{-1}(\theta)\|_F \leq \frac{\|\theta - z\|_{\Psi(\theta)}}{(1 - \|\theta - z\|_{\Psi(\theta)})^2} \qquad \forall \theta, z \in \mathrm{Int}(K). \tag{5}$$

Laddha et al. (2020) show that (5) holds in the special case when $\Phi$ is replaced with the Hessian $H(\theta)$ of the log-barrier function without a regularizer. However, the soft-threshold log-barrier Hessian, $\Phi$, is not the Hessian of a log-barrier function for any set of equations defining the polytope $K$. To get around this problem, we instead use the fact (from Mangoubi & Vishnoi (2022)) that $\Phi$ is the limit of a sequence of matrices $\{H_j(\theta)\}_{j=1}^\infty$ where each $H_j$ is the Hessian of a log-barrier function for an increasing set of (redundant) inequalities defining the polytope $K$. Combining these two facts

allows us to show Equation (5) holds for the Hessian $\Phi$ of the log-barrier function with soft-threshold regularizer (Lemma B.1). Next, we apply (5) to show that (Lemma B.2),

$$\|(\hat{S}^2(z) - \hat{S}^2(\theta))\hat{S}^{-2}(\theta)\|_F \leq \|(\Psi(z) - \Psi(\theta))^{-1}\Psi^{-1}(\theta)\|_F \leq \frac{\alpha^{-\frac{1}{2}}\|\theta - z\|_{\Phi(\theta)}}{(1 - \alpha^{-\frac{1}{2}}\|\theta - z\|_{\Phi(\theta)})^2} \leq O(\alpha^{-\frac{1}{2}}\sqrt{d}),$$

w.h.p. As the step size $\alpha = \Theta(1/d)$, we get that $\|(\hat{S}^2(z) - \hat{S}^2(\theta))\hat{S}^{-2}(\theta)\|_F \leq O(1)$. w.h.p.

**Computing a randomized estimate for the determinantal term.** To compute an estimate for the determinantal term in the Metropolis acceptance rule, we use a well-known method for estimating matrix-valued functions using polynomials (see e.g. Barvinok (2017) Rudelson & Zeitouni (2016)). A related approach to the one presented here for computing the determinant in the special case when $f \equiv 0$ was given in Laddha et al. (2020), however it contains a number of gaps in the proof and algorithm. We discuss these gaps, and the differences between their approach and the one given here, in Section C.

Towards this, one can apply the linear solver for $\Phi(\theta)$ to compute a random estimate $Y$ with mean $\log\det(\Phi(\theta)) - \log\det(\Phi(z))$: $Y = v^\top(\hat{A}^\top(t\hat{S}(\theta)^{-2} + (1-t)I_{m+d})\hat{A})^{-1}\hat{A}^\top(\hat{S}(\theta)^{-2} - I_{m+d})\hat{A}v + \log\det\hat{A}^\top\hat{A}$, Where $v \sim N(0, I_d)$ and $t \sim \text{uniform}(0,1)$. While this provides an estimate for the log-determinant of $\Phi$, it does not give an estimate with mean equal to a Metropolis acceptance rule for $\pi$.

We first replace the Metropolis acceptance rule $\min(\frac{p(z \to \theta)}{p(\theta \to z)}e^{f(z) - f(\theta)}, 1) = \min\left(\frac{\sqrt{\det(\Phi(z))}}{\sqrt{\det(\Phi(\theta))}}e^{\|z - \theta\|_{\Phi(\theta)}^2 - \|\theta - z\|_{\Phi(z)}^2}e^{f(\theta) - f(z)}, 1\right)$ with a (different) Metropolis acceptance rule, $\mathsf{a}(\theta, z) := \left(1 + (\frac{\det(\Phi(z))}{\det(\Phi(\theta))})^{-\frac{1}{2}}\right)^{-1} \times \min\left(\frac{e^{-f(z)}}{e^{-f(\theta)}} \times e^{\|z - \theta\|_{\Phi(\theta)}^2 - \|\theta - z\|_{\Phi(z)}^2}, 1\right)$, whose determinantal term is a smooth function of $\det(\Phi(\theta))$ and $\det(\Phi(z))$:

$$\left(1 + (\tfrac{\det(\Phi(z))}{\det(\Phi(\theta))})^{-\frac{1}{2}}\right)^{-1} = \text{sigmoid}\left(\tfrac{1}{2}\log\det(\Phi(z)) - \tfrac{1}{2}\log\det(\Phi(\theta))\right), \tag{6}$$

where $\text{sigmoid}(t) := \frac{1}{1 + e^{-t}}$. Since $\mathsf{a}(\theta, z)p(\theta \to z)\pi(\theta) = \mathsf{a}(z, \theta)p(z \to \theta)\pi(z)$, this acceptance rule preserves the target stationary distribution $\pi(\theta) \propto e^{-f(\theta)}$ of the Markov chain.

Next, we would ideally like to plug i.i.d. samples $Y_1, \cdots, Y_n$ of mean $\log\det\Phi(\theta) - \log\det\Phi(z)$ into a Taylor series for $\text{sigmoid}(t)$ to obtain an estimate with mean equal to the l.h.s. of (6). Unfortunately, the Taylor series of $\text{sigmoid}(t)$ at 0 has finite region of convergence $(-\pi, \pi)$.

While one can use convexity of the function $\log\det(\Phi(\theta))$, together with a bound on the gradient of $\log\det(\Phi(\theta))$, to show that $\frac{1}{2}\log\det(\Phi(z)) - \frac{1}{2}\log\det(\Phi(\theta)) > -\pi$ w.h.p., one may have that $\frac{1}{2}\log\det(\Phi(z)) - \frac{1}{2}\log\det(\Phi(\theta)) > \pi$ with probability $\Omega(1)$.

To get around this problem, we use two different series expansions for $\text{sigmoid}(t)$ with overlapping regions of convergence:

1. a Taylor series expansion centered at 0 with region of convergence $(-\pi, \pi)$,
2. a series expansion "at $+\infty$" with region of convergence $(0, +\infty)$, which is a polynomial in $e^{-t}$: $\text{sigmoid}(t) = \sum_{k=0}^\infty c_k e^{-kt}$.

To determine which series expansion to use at each step of the Dikin walk, we show that each random estimate $Y$ concentrates in the ball of radius $\frac{1}{8}$ about its mean $\mathbb{E}[Y] = \log\det(\Phi(z)) - \log\det(\Phi(\theta))$ w.h.p. (Lemma B.3). As the regions of convergence of the two series expansions have intersection $(0, \pi)$, this allows us to use the value of the random estimate $Y$ to select a choice of series expansion that, with high probability, contains $\mathbb{E}[Y] = \log\det(\Phi(z)) - \log\det(\Phi(\theta))$.

**Bounding the number of arithmetic operations.** We have already shown that computing the proposed update $z$ (Line 6 of Algorithm 1) and computing each random estimate $Y_j$ for the log-determinant (Line 14) can be done in $O(\text{nnz}(A) + d^2)$ arithmetic operations at each step of the Markov chain by using the efficient inverse maintenance linear solver.

As both series expansions converge exponentially fast in the number of terms, only $\log(1/\delta)$ terms in the series expansion are needed to compute an estimate $X$ (Lines 18 and 21) with

mean within error $O(\delta)$ of the determinantal term in the Metropolis acceptance probability $\left| \mathbb{E}[X] - \left( 1 + \left( \frac{\det(\Phi(z))}{\det(\Phi(\theta))} \right)^{-\frac{1}{2}} \right)^{-1} \right| \leq O(\delta)$. Thus, the number of calls to the linear solver at each Markov chain step is $O(\log(1/\delta))$, taking $O((\mathrm{nnz}(A) + d^2) \log(1/\delta))$ arithmetic operations.

Computing the terms $f(\theta)$ and $f(z)$ in the Metropolis acceptance rule requires two calls to the oracle for $f$, which takes $O(T_f)$ arithmetic operations, at each step of the Markov chain. The other steps in the Algorithm take no more time to compute. In particular, while several "initialization" steps for the linear solver and other computations (Lines 4-6) require $O(md^{\omega-1})$ arithmetic operations, these computations are not repeated at each step of the Markov chain, and thus do not change the overall runtime.

Since the Markov chain is run for roughly $T = O((md + L^2R^2) \log(w/\delta))$ steps, the runtime is roughly $O(T((\mathrm{nnz}(A) + d^2) \log(1/\delta + T_f)) = O((md + L^2R^2)(\mathrm{nnz}(A) + d^2 + T_f) \log^2(w/\delta)$.

**Bounding the total variation distance.** Mangoubi & Vishnoi (2022) show that the soft-threshold Dikin walk outputs a point with distribution $\mu_T$ within TV distance $O(\delta)$ of the target distribution $\pi$ after roughly $T = O((md + L^2R^2) \log(w/\delta))$ steps; the same TV bound holds if one replaces their acceptance rule with the smooth acceptance rule we use.

Our bound on the TV distance follows directly from their bound, with some minor adjustments. This is because, while we give a more efficient algorithm to *implement* the soft-threshold Dikin walk Markov chain, the Markov chain itself is (approximately) the same as the Markov chain considered in their work. The main difference (aside from our use of a smoothed Metropolis acceptance rule) is that the (mean of) the randomized estimate we compute for the acceptance probability is accurate to within some small error $O(\delta)$, since we only compute a finite number of terms in the Taylor series.

Thus, we can define a coupling between the Markov chain $\theta_1, \theta_2, \ldots, \theta_T$ generated by our algorithm and the Markov chain $\mathcal{Y}_1, \mathcal{Y}_2, \ldots \mathcal{Y}_T$ defined in their paper (with smoothed acceptance rule), such that $\theta_i = \mathcal{Y}_i$ at every $i \in \{1, \cdots, T\}$ with probability $\geq 1 - T\delta$. This implies that the distribution $\mu$ of the last step of our Markov chain is within total variation distance $O(T\delta)$ of the distribution $\nu_T$ of $\mathcal{Y}_T$. Thus, $\|\mu - \pi\|_{\mathrm{TV}} \leq \|\mu - \nu_T\|_{\mathrm{TV}} + \|\nu_T - \pi\|_{\mathrm{TV}} \leq O(T\delta) + O(\delta)$. Pluggining in $\delta/T$ in place of $\delta$, we get $\|\mu - \pi\|_{\mathrm{TV}} \leq \delta$.

## 5 CONCLUSION

Our main result improves on the runtime bounds of a line of previous works for the problem of sampling from a log-Lipschitz or log-smooth log-concave distribution on a polytope (Table 1). Key to our result is showing that the regularized log-barrier Hessian changes slowly at each step of the soft-threshold Dikin walk, which allows us to deploy fast linear solvers from the interior point method literature to reduce the per-iteration complexity of the walk.

The use of fast linear solvers allows us to achieve a per-iteration complexity $\mathrm{nnz}(A) + d^2$ for the soft-threshold Dikin walk that is nearly-linear in the input complexity of the problem when $A$ is a dense matrix; when $A$ is sparse, the dependence of the per-iteration complexity on $\mathrm{nnz}(A)$ is nearly-linear as well. On the other hand, it would be interesting to see if the $d^2$ term can removed by deploying a different choice of fast linear solver for the Hessian.

Finally, while the fastest-known algorithms for minimizing linear functions on a polytope $K$ have runtime (nearly) equal to the matrix multiplication time Cohen et al. (2021); Jiang et al. (2021), the best runtime bounds for sampling from a distribution $\pi \propto e^{-f}$ on a polytope (even when $f$ is uniform) are a larger polynomial in the dimension $d$. Thus, it would be interesting to see if the overall runtime (i.e., number of steps times the per-step complexity) of sampling methods can be improved to match the runtime of linear programming methods.

### ACKNOWLEDGMENTS

NV was supported in part by an NSF CCF-2112665 award. OM was supported in part by an NSF CCF-2104528 award and a Google Research Scholar award. The authors thank Yin Tat Lee and Sushant Sachdeva for valuable feedback and discussions.

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

Contents

## A   PRELIMINARIES

The following lemma of Lee & Sidford (2015) allows us to maintain an efficient linear systems solver for the Hessian of the regularized barrier function $\Phi$:

**Lemma A.1 (Efficient inverse maintenance, Theorem 13 in Lee & Sidford (2015),)** *Suppose that a sequence of matrices $A^\top C^{(k)} A$ for the inverse maintenance problem, where the sequence of diagonal matrices $C^{(1)}, C^{(1)} \ldots$, with $C^{(k)} = \operatorname{diag}(c_1^{(k)}, \ldots, c_d^{(k)})$, satisfies $\sum_{i=1}^d \left( \frac{c_i^{(k+1)} - c_i^{(k)}}{c_i^{(k)}} \right)^2 = O(1)$ for all $k \in \mathbb{N}$. Then there is an algorithm that maintains a $\tilde{O}(\operatorname{nnz}(A) + d^2)$ arithmetic operation linear system solver for the sequence of matrices $\{A^\top C^{(k)} A\}_{k=1}^T$ for $T$ rounds in a total of $\tilde{O}(T(\operatorname{nnz}(A) + d^2) + d^\omega)$ arithmetic operations over all rounds $T$ rounds.*

For any symmetric positive definite matrix $M \in \mathbb{R}^{d \times d}$, define $\|h\|_M := h^\top M h$.

The following lemma is useful in bounding the change in the log-barrier Hessian:

**Lemma A.2 (Lemmas 1.2 & 1.5 in Laddha et al. (2020))** *Let $K := \{\theta \in \mathbb{R}^d : A\theta \leq b\}$, where $A \in \mathbb{R}^{m \times d}$ and $b \in \mathbb{R}^m$ such that $K$ is bounded with non-empty interior. Let $H(\theta) = \sum_{j=1}^m \frac{a_j a_j^\top}{(b_j - a_j^\top \theta)^2}$ be the Hessian of the log-barrier function for $K$. Then for any $x, y \in \operatorname{Int}(K)$ with $\|x - y\|_{H(x)} < 1$, we have $\|H^{-\frac{1}{2}}(x)(H(y) - H(x))H^{-\frac{1}{2}}(x)\|_F \leq \frac{\|x-y\|_{H(x)}}{(1-\|x-y\|_{H(x)})^2}$.*

The following lemma gives a high-probability *lower* bound on the determinantal term $\frac{\det(\Phi(z))}{\det(\Phi(\theta))}$ in the Metropolis acceptance ratio.

**Lemma A.3 (Lemma 6.9 in Mangoubi & Vishnoi (2022))** *Consider any $\theta \in \operatorname{Int}(K)$, and $\xi \sim N(0, I_d)$. Let $z = \theta + (\Phi(\theta))^{-\frac{1}{2}} \xi$. Then*

$$\mathbb{P}\left( \frac{\det(\Phi(z))}{\det(\Phi(\theta))} \geq \frac{48}{50} \right) \geq 1 - \frac{98}{100}, \tag{7}$$

*and*

$$\mathbb{P}\left( \|z - \theta\|_{\Phi(z)}^2 - \|z - \theta\|_{\Phi(\theta)}^2 \leq \frac{2}{50} \right) \geq \frac{98}{100}. \tag{8}$$

The following lemma of Mangoubi & Vishnoi (2022) says that the Hessian $\Phi$ of the regularized barrier function, while not the Hessian of a log-barrier function for any system of inequalities defining $K$, is nevertheless the limit of an infinite sequence of matrices $H_j$, where each $H_j$ the Hessian of a logarithmic barrier function for a system of inequalities defining the same polytope $K$.

**Lemma A.4 (Lemma 6.7 in Mangoubi & Vishnoi (2022))** *There exits a sequence of matrix-valued functions $\{H_j\}_{j=1}^\infty$, where each $H_j$ is the Hessian of a log-barrier function on $K$, such that for all $w \in \operatorname{Int}(K)$,*

$$\lim_{j \to \infty} H_j(w) = \alpha\Phi(w), \qquad \text{and} \qquad \lim_{j \to \infty} (H_j(w))^{-1} = \alpha^{-1}(\Phi(w))^{-1},$$

*uniformly in $w$.*

The following Lemma (Lemma A.5) gives a mixing time bound for a basic implementation of the soft-threshold Dikin walk (Algorithm 2, below):

---

**Algorithm 2:** Basic implementation of soft-threshold Dikin walk

---

**Input:** $m, d \in \mathbb{N}$, $A \in \mathbb{R}^{m \times d}$, $b \in \mathbb{R}^m$, which define the polytope $K := \{\theta \in \mathbb{R}^d : A\theta \leq b\}$
**Input:** $\mathcal{Y}_0 \in \text{Int}(K)$, $T \in \mathbb{N}$

1   **Hyperparameters:** $\alpha > 0$; $\eta > 0$; $T \in \mathbb{N}$;
2   **for** $i = 0, \ldots, T-1$ **do**
3      Sample $\zeta_i \sim N(0, I_d)$
4      Set $H(\mathcal{Y}_i) = \sum_{j=1}^m \frac{a_j a_j^\top}{(b_j - a_j^\top \mathcal{Y}_i)^2}$
5      Set $\Phi(\mathcal{Y}_i) = \alpha^{-1} H(\mathcal{Y}_i) + \eta^{-1} I_d$
6      Set $\mathcal{Z}_i = \Phi(\mathcal{Y}_i)^{-\frac{1}{2}} \zeta_i$
7      **if** $\mathcal{Z}_i \in \text{Int}(K)$ **then**
8          Set $H(\mathcal{Z}_i) = \sum_{j=1}^m \frac{a_j a_j^\top}{(b_j - a_j^\top \mathcal{Z}_i)^2}$
9          Set $\Phi(\mathcal{Z}_i) = \alpha^{-1} H(\mathcal{Y}_i) + \eta^{-1} I_d$
10          Set $p_i = \frac{1}{2} \frac{\frac{\det(\Phi(\mathcal{Z}_i))}{\det(\Phi(\mathcal{Y}_i))}}{\frac{\det(\Phi(\mathcal{Y}_i))}{\det(\Phi(\mathcal{Z}_i))} + \frac{\det(\Phi(\mathcal{Z}_i))}{\det(\Phi(\mathcal{Y}_i))}} \times \min\left( \frac{e^{-f(\mathcal{Z}_i)}}{e^{-f(\mathcal{Y}_i)}} \times e^{\|\mathcal{Z}_i - \mathcal{Y}_i\|_{\Phi(\mathcal{Y}_i)}^2 - \|\mathcal{Y}_i - \mathcal{Z}_i\|_{\Phi(\mathcal{Z}_i)}^2}, \ 1 \right)$
11          Set $\mathcal{Y}_{i+1} = \mathcal{Z}_i$ with probability $p_i$,
12      **else**
13          Reject $\mathcal{Z}_i$.
14      **end**
15      **Output:** $\mathcal{Y}_T$
16   **end**

---

**Lemma A.5 (Lemma 6.15 of Mangoubi & Vishnoi (2022))** *Let $\mathcal{Y}_0, \mathcal{Y}_1, \cdots, \mathcal{Y}_T$ be the Markov chain generated by Algorithm 2. Let $\delta > 0$. Suppose that $f : K \to \mathbb{R}$ is either $L$-Lipschitz (or has $\beta$-Lipschitz gradient). Suppose that $\mathcal{Y}_0 \sim \nu_0$ where $\nu_0$ is a $w$-warm distribution with respect to $\pi \propto e^{-f}$ with support on $K$. For any $t > 0$ denote by $\nu_t$ denote the distribution of $\mathcal{Y}_t$, for $\alpha \leq \frac{1}{10^5 d}$ and $\eta \leq \frac{1}{10^4 dL^2}$ (or $\eta \leq \frac{1}{10^4 d\beta}$). Then for any $T \geq 10^9 \left(2m\alpha^{-1} + \eta^{-1} R^2\right) \times \log(\frac{w}{\delta})$ we have that*

$$\|\nu_T - \pi\|_{\text{TV}} \leq \delta.$$

Note: Lemma A.5 is given in Mangoubi & Vishnoi (2022) for the same Markov chain as in Algorithm 2, but with acceptance rule $\frac{1}{2} \times \min\left( \frac{e^{-f(\mathcal{Z}_i)}\sqrt{\det(\Phi(\mathcal{Z}_i))}}{e^{-f(\mathcal{Y}_i)}\sqrt{\det(\Phi(\mathcal{Y}_i))}} \times e^{\|\mathcal{Z}_i - \mathcal{Y}_i\|_{\Phi(\mathcal{Y}_i)}^2 - \|\mathcal{Y}_i - \mathcal{Z}_i\|_{\Phi(\mathcal{Z}_i)}^2}, \ 1 \right)$ in place of the acceptance rule $p_i$ used in Algorithm 2. The same proof of Lemma A.5 given in Mangoubi & Vishnoi (2022) holds for the above acceptance rule in Algorithm 2 with minor modifications.

The following trace inequality is needed in the proofs.

**Lemma A.6 (von Neumann trace Inequality Von Neumann (1962))** *Let $M, Z \in \mathbb{R}^{d \times d}$ be matrices, and denote by $\sigma_1 \geq \cdots \geq \sigma_d$ the singular values of $M$ and by $\gamma_1 \geq \cdots \geq \gamma_d$ the singular values of $Z$. Then*

$$|\text{tr}(MZ)| \leq \sum_{i=1}^d \sigma_i \gamma_i.$$

We need the following fact.

**Proposition A.7** *Let $D \in \mathbb{R}^{(d+m) \times (d+m)}$ be a diagonal matrix. Then*

$$\log \det(\hat{A}^\top D \hat{A}) - \log \det(\hat{A}^\top \hat{A}) = \int_0^1 \text{tr}\left( \tau \hat{A}^\top D \hat{A} + (1-\tau)\hat{A}^\top \hat{A})^{-1} (\hat{A}^\top D \hat{A} - \hat{A}^\top \hat{A}) \right) d\tau.$$

**Proof:**

Let $\Theta(\tau) := \tau \hat{A}^\top D \hat{A} + (1-\tau) \hat{A}^\top \hat{A}$ for any $\tau \geq 0$. Then

$$\frac{\mathrm{d}}{\mathrm{d}\tau} \log \det \Theta(\tau) = \mathrm{tr}\left(\frac{\mathrm{d}}{\mathrm{d}\tau} \log(\Theta(\tau))\right) = \mathrm{tr}\left(\Theta^{-1}(\tau) \frac{\mathrm{d}}{\mathrm{d}\tau} \Theta(\tau)\right)$$

$$= \mathrm{tr}\left((\tau \hat{A}^\top D \hat{A} + (1-\tau)\hat{A}^\top \hat{A})^{-1}(\hat{A}^\top D \hat{A} - \hat{A}^\top \hat{A})\right).$$

Thus,

$$\log \det(\hat{A}^\top D \hat{A}) - \log \det(\hat{A}^\top \hat{A}) = \int_0^1 \frac{\mathrm{d}}{\mathrm{d}\tau} \log \det \Theta(\tau) \mathrm{d}\tau$$

$$= \int_0^1 \mathrm{tr}\left((\tau \hat{A}^\top D \hat{A} + (1-\tau)\hat{A}^\top \hat{A})^{-1}(\hat{A}^\top D \hat{A} - \hat{A}^\top \hat{A})\right) \mathrm{d}\tau.$$

∎

Finally, we need the following concentration inequality to show a high probability bound on the error $Y - \mathbb{E}[Y]$ of the estimator $Y$ for $\log \det \Phi(z) - \log \det \Phi(\theta)$.

**Lemma A.8 (Hanson-Wright concentration inequality)** *Let $M \in \mathbb{R}^{d \times d}$. Let $v \sim N(0, I_d)$. Then for all $t > 0$, $\mathbb{P}(|v^\top M v| > t) \leq 2 \exp\left(-8 \min\left(\frac{t^2}{\|M\|_F^2}, \frac{t}{\|M\|_2}\right)\right).$*

## B  PROOF OF THE MAIN RESULT

To prove Theorem 2.1, we first show that the Frobenius norm of the Hessian of the log-barrier function with soft-threshold regularizer changes slowly with respect to the local norm (Lemmas B.1 and B.2). We also prove a concentration inequality for the random estimate $Y$ for the log-determinant (Lemma B.3).

We then give a proof of Theorem 2.1. The first part of the proof of Theorem 2.1 bounds the running time, by using Lemma B.2 together with Lemma A.1 in the preliminaries to show that one can deploy the fast linear solver to obtain a bound of $\mathrm{nnz}(A) + d^2$ on the per-iteration complexity of Algorithm 1.

The second part of the proof bounds the total variation distance between the output of Algorithm 1 and the target distribution $\pi$. Towards this, we use the concentration bound in Lemma B.3 to show that the series expansions in Lines 18 and 21 of Algorithm 1 converge w.h.p. and give a good estimate for the Metropolis acceptance probability. To conclude, use this fact to show that there exists a coupling between the Markov chain computed by Algorithm 1 and an "exact" implementation of the soft-barrier Dikin walk Markov such that these two coupled Markov chains are equal w.h.p. This allows us to use the bound on the total variation for the "exact" Markov chain (Lemma A.5 in the preliminaries) to obtain a bound on the total variation error of the output of Algorithm 1.

**Lemma B.1** *Let $\varphi(\theta) := -\sum_{j=1}^m \log(b_j - a_j^\top \theta)$ be a log-barrier function for $K := \{\theta \in \mathbb{R}^d : A\theta \leq b\}$, where $A \in \mathbb{R}^{m \times d}$ and $b \in \mathbb{R}^m$. Let $c \geq 0$, and define the matrix-valued function $\Psi(\theta) := \nabla^2 \varphi(\theta) + c I_d$. Then for any $\theta, z \in \mathrm{Int}(K)$ with $\|\theta - z\|_{\Psi(\theta)} < 1$, we have $\|\Psi^{-\frac{1}{2}}(\theta)(\Psi(z) - \Psi(\theta))\Psi^{-\frac{1}{2}}(\theta)\|_F \leq \frac{\|\theta - z\|_{\Psi(\theta)}}{(1 - \|\theta - z\|_{\Psi(\theta)})^2}.$*

**Proof:**

Let $\theta, z \in \mathrm{Int}(K)$ be any two points in $\mathrm{Int}(K)$ such that $\|\theta - z\|_{\Psi(\theta)} < 1$.

From Lemma A.4, we have that there exits a sequence of matrix functions $\{H_j\}_{j=1}^\infty$, where each matrix is the Hessian of a log-barrier function on $K$, such that for all $w \in \mathrm{Int}(K)$,

$$\lim_{j \to \infty} H_j(w) = \Phi(w), \tag{9}$$

uniformly in $w$, and

$$\lim_{j \to \infty} (H_j(w))^{-1} = (\Phi(w))^{-1}, \tag{10}$$

uniformly in $w$.

Thus we have

$$\lim_{j \to \infty} \|\theta - z\|_{H_j(\theta)} = \lim_{j \to \infty} \sqrt{(\theta - z)^\top H_j(\theta)(\theta - z)}$$

$$= \sqrt{(\theta - z)^\top \lim_{j \to \infty} H_j(\theta)(\theta - z)}$$

$$\overset{\text{Eq. (9)}}{=} \sqrt{(\theta - z)^\top \Psi(\theta)(\theta - z)}$$

$$= \|\theta - z\|_{\Psi(\theta)}. \tag{11}$$

In particular, since $\|\theta - z\|_{\Psi(\theta)} < 1$, (11) implies that there exists $J \in \mathbb{N}$ such that

$$\|\theta - z\|_{H_j(\theta)} < 1 \qquad \forall j \geq J. \tag{12}$$

Moreover, since $H_j(w)$ is the Hessian of a log-barrier function on $K$ for every $j \in \mathbb{N}$, by Lemma A.2 and (12) we have that

$$\|H_j^{-\frac{1}{2}}(\theta)(H_j(z) - H_j(\theta))H_j^{-\frac{1}{2}}(\theta)\|_F \leq \frac{\|\theta - z\|_{H_j(\theta)}}{(1 - \|\theta - z\|_{H_j(\theta)})^2} \qquad \forall j \geq J. \tag{13}$$

Moreover,

$$\lim_{j \to \infty} \|H_j^{-\frac{1}{2}}(\theta)(H_j(z) - H_j(\theta))H_j^{-\frac{1}{2}}(\theta)\|_F$$

$$\overset{\text{Eq. (13)}}{=} \|(\lim_{j \to \infty} H_j^{-\frac{1}{2}}(\theta))(\lim_{j \to \infty} H_j(z) - \lim_{j \to \infty} H_j(\theta))(\lim_{j \to \infty} H_j^{-\frac{1}{2}}(\theta))\|_F$$

$$\overset{\text{Eq. (9),(10)}}{=} \|\Psi^{-\frac{1}{2}}(\theta)(\Psi(z) - \Psi(\theta))\Psi^{-\frac{1}{2}}(\theta)\|_F \tag{14}$$

Therefore, plugging (14) and (11) into (13), we have that

$$\|\Psi^{-\frac{1}{2}}(\theta)(\Psi(z) - \Psi(\theta))\Psi^{-\frac{1}{2}}(\theta)\|_F \leq \frac{\|x - y\|_{\Psi(\theta)}}{(1 - \|\theta - z\|_{\Psi(\theta)})^2}.$$

∎

**Lemma B.2** *Let $z = \theta + \Phi(\theta)^{-\frac{1}{2}}\xi$ where $\xi \sim N(0, I_d)$. Then we have with probability at least $1 - \gamma$ that*

$$\|\hat{S}(\theta)^{-2}\hat{S}(z)^2 - I_m\|_F \leq \frac{1}{1000 \log(\frac{1}{\gamma})}.$$

**Proof:** Setting $\Psi(\theta) = \hat{A}^\top \hat{S}(\theta)^2 \hat{A}$, we have that $\Psi(\theta) = \alpha\Phi(\theta) = H(\theta) + \alpha\eta^{-1}I_d$.

Hence, we have by Lemma B.1 that, with probability at least $1 - \gamma$,

$$
\begin{aligned}
\|\hat{S}(\theta)^{-2}\hat{S}(z)^2 - I_m\|_F &= \|\hat{S}(\theta)^{-1}\hat{S}(z)^2\hat{S}(\theta)^{-1} - I_m\|_F \\
&\le \|(\hat{A}^\top \hat{S}(\theta)^2\hat{A})^{-\frac{1}{2}}\hat{A}^\top\hat{S}(z)^2\hat{A}(\hat{A}^\top\hat{S}(\theta)^2\hat{A})^{-\frac{1}{2}} - I_d\|_F \\
&= \|\Phi^{-\frac{1}{2}}(x)\Phi(y)\Phi^{-\frac{1}{2}}(x) - I_d\|_F \\
&= \|\Phi^{-\frac{1}{2}}(x)(\Phi(y) - \Phi(x))\Phi^{-\frac{1}{2}}(x)\|_F \\
&= \|\alpha^{-\frac{1}{2}}\Phi^{-\frac{1}{2}}(x)(\alpha\Phi(y) - \alpha\Phi(x))\alpha^{-\frac{1}{2}}\Phi^{-\frac{1}{2}}(x)\|_F \\
&= \|\Psi^{-\frac{1}{2}}(x)(\Psi(y) - \Psi(x))\Psi^{-\frac{1}{2}}(x)\|_F \\
&\le \frac{\|x - y\|_{\Psi(x)}}{(1 - \|x - y\|_{\Psi(x)})^2} \\
&= \frac{\alpha^{-\frac{1}{2}}\|\theta - z\|_{\Phi(\theta)}}{(1 - \alpha^{-\frac{1}{2}}\|\theta - z\|_{\Phi(\theta)})^2} \\
&\le \alpha^{-\frac{1}{2}}8\sqrt{d}\log^{\frac{1}{2}}\left(\frac{1}{\gamma}\right) \\
&\le \frac{1}{1000},
\end{aligned}
$$

where the second inequality holds by Lemma B.1. The fourth inequality holds because $\alpha = \frac{1}{10^5 d}\log^{-1}\left(\frac{1}{\gamma}\right)$. The third inequality holds with probability at least $1 - \gamma$ because

$$
\|\theta - z\|_{\Phi(\theta)}^2 = (\theta - z)^\top\Phi(\theta)(\theta - z) = (\Phi(\theta)^{-\frac{1}{2}}\xi)^\top\Phi(\theta)(\Phi(\theta)^{-\frac{1}{2}}\xi) \le 8\sqrt{d}\log(\frac{1}{\gamma})
$$

with probability at least $1 - \gamma$ by the Hanson-Wright inequality (Lemma A.8). ∎

**Lemma B.3** *Let $W \in \mathbb{R}^{m+d \times m+d}$ be a diagonal matrix, and let $t > 0$. Define $\Theta(t) := \hat{A}^\top(I_{m+d} + t(W - I_{m+d}))\hat{A}$. Let $v \sim N(0, I_d)$, and let*

$$
Y = v^\top\Theta(t)^{-1}\hat{A}^\top(W - I_m)\hat{A}v + \log\det\hat{A}^\top\hat{A}. \tag{15}
$$

*Suppose that $\frac{1}{2}I_m \preceq W \preceq 2I_m$ and that $\|W - I_m\|_F \le c$ for some $c > 0$. Then*

$$
\mathbb{P}(|Y - \mathbb{E}[Y]| \ge s \cdot c) \le e^{-\frac{1}{8}s} \qquad \forall s \ge 0. \tag{16}
$$

**Proof:**

$$
\begin{aligned}
\|\Theta(t)^{-1}\hat{A}^\top(W - I_m)\hat{A}\|_F &\le \|(\hat{A}^\top(tW + (1-t)I_m)\hat{A})^{-1}\|_2 \cdot \|A^\top(W - I_m)A\|_F \\
&\le \|(\hat{A}^\top(\tfrac{1}{2}I_m)\hat{A})^{-1}\|_2 \cdot \|\hat{A}^\top(W - I_m)\hat{A}\|_F \\
&= 2\|(\hat{A}^\top\hat{A})^{-1}\|_2 \cdot \|\hat{A}^\top(W - I_m)\hat{A}\|_F \\
&= 2\|(\hat{A}^\top\hat{A})^{-1}\|_2 \cdot \mathrm{Tr}^{\frac{1}{2}}((\hat{A}^\top(W - I_m)\hat{A})^2) \\
&= 2\|(\hat{A}^\top\hat{A})^{-1}\|_2 \cdot \mathrm{Tr}^{\frac{1}{2}}(\hat{A}^\top(W - I_m)\hat{A}\hat{A}^\top(W - I_m)\hat{A})\hat{A})^{-1}\|_2 \\
&\qquad \cdot \mathrm{Tr}^{\frac{1}{2}}(\hat{A}\hat{A}^\top(W - I_m)\hat{A}\hat{A}^\top(W - I_m)) \\
&\le 2\|(\hat{A}^\top\hat{A})^{-1}\|_2 \cdot \sqrt{\|\hat{A}\hat{A}^\top\|_2} \cdot |\mathrm{Tr}^{\frac{1}{2}}((W - I_m)\hat{A}\hat{A}^\top(W - I_m))| \\
&\le 2\|(\hat{A}^\top\hat{A})^{-1}\|_2 \cdot \sqrt{\|\hat{A}\hat{A}^\top\|_2} \cdot |\mathrm{Tr}^{\frac{1}{2}}(\hat{A}\hat{A}^\top(W - I_m)^2)| \\
&\le 2\|(\hat{A}^\top\hat{A})^{-1}\|_2 \cdot \|\hat{A}\hat{A}^\top\|_2 \cdot |\mathrm{Tr}^{\frac{1}{2}}((W - I_m)^2)| \\
&= 2\|W - I_m\|_F, \\
&\le 2c, \tag{17}
\end{aligned}
$$

where the fourth and fifth inequalities hold by the Von Neumann trace inequality (Lemma A.6). Thus, by the Hanson-Wright inequality (Lemma A.8) we have that for every $s \geq 0$,

$$\mathbb{P}(|v^\top \Theta(t)^{-1}\hat{A}^\top(W - I_m)\hat{A}v - \mathbb{E}[v^\top \Theta(t)^{-1}\hat{A}^\top(W - I_m)\hat{A}v]| \geq s)$$

$$\leq 2\exp\left(-\frac{1}{8}\min\left(\frac{s^2}{\|\Theta(t)^{-1}\hat{A}^\top(W - I_m)\hat{A}\|_F^2}, \frac{s}{\|\Theta(t)^{-1}\hat{A}^\top(W - I_m)\hat{A}\|_2}\right)\right)$$

$$\leq 2\exp\left(-\frac{1}{8}\frac{s}{\|\Theta(t)^{-1}\hat{A}^\top(W - I_m)\hat{A}\|_F}\right). \tag{18}$$

Plugging (17) into (18), we get that,

$$\mathbb{P}(|v^\top \Theta(t)^{-1}\hat{A}^\top(W - I_m)\hat{A}v - \mathbb{E}[v^\top \Theta(t)^{-1}\hat{A}^\top(W - I_m)\hat{A}v]| \geq sc) \leq 2e^{-\frac{1}{8}s} \qquad \forall s \geq 0.$$

$\blacksquare$

**Proof:** [of Theorem 2.1]

**Bounding the runtime:**

*Cost of maintaining and using a linear solver:* By Lemma B.2, we have that at every iteration of the outer "for" loop in Algorithm 1,

$$\sum_{i=1}^d \left(\frac{\hat{S}(z)^2[i] - \hat{S}(\theta)^2[i]}{\hat{S}(\theta)^2[i]}\right)^2 = \|\hat{S}(\theta)^{-2}\hat{S}(z)^2 - I_m\|_F^2 \leq O(1), \tag{19}$$

with probability at least $1 - \gamma$.

Rewriting the l.h.s. of (19), we also have that with probability at least $1 - \gamma$,

$$\sum_{i=1}^d \left((t\hat{S}(z)^2[i]\hat{S}(\theta)^{-2}[i] + (1-t)) - 1\right)^2 \leq \sum_{i=1}^d \left(\hat{S}(z)^2[i]\hat{S}(\theta)^{-2}[i] - 1\right)^2 \leq O(1), \tag{20}$$

for any $t \in [0, 1]$.

To implement Algorithm 1, we use efficient inverse maintenance to maintain linear solvers for $\Phi(\theta) = A^\top\hat{S}(\theta)^{-2}A$ and $\hat{A}^\top\Psi(\hat{S}(\theta)^{-2}\hat{S}(z)^2, t)\hat{A} = \hat{A}^\top(t\hat{S}(\theta)^{-2}\hat{S}(z)^2 + (1 - t)I_{m+d})\hat{A}$ for $T$ rounds, where $T$ is the number of Dikin walk steps. Plugging Inequality (19) (for the $\Phi(\theta)$ linear system solver) and Inequality (20) (for the $\hat{A}^\top\Psi(\hat{S}(\theta)^{-2}\hat{S}(z)^2, t)\hat{A}$ linear system solver) into Lemma A.1, we get that maintaining these linear system solvers for $T$ steps (Lines 2, 26, 2, and 12 in Algorithm 1), and applying each linear solver at most $\mathcal{N}$ times at each step (Lines 6 and 14), takes $O(T\mathcal{N}(\text{nnz}(A) + d^2) + d^\omega)$ arithmetic operations.

Since $T \geq d^{\omega-2}$, we have that the number of arithmetic operations is $O(T\mathcal{N}(\text{nnz}(A) + d^2))$.

*Cost of other steps:* The remaining steps require less that $O(T\mathcal{N}(\text{nnz}(A) + d^2))$ steps to compute:

- Each time they are run, Lines 5 and 13 can be accomplished in $O(\text{nnz}(A))$ arithmetic operations by standard matrix-vector multiplication. Sampling a $d$-dimensional Gaussian vector (Lines 4 and 10) can be done in $O(d) \leq O(\text{nnz}(A))$ arithmetic operations and sampling from the uniform distribution 11 takes $O(1)$ arithmetic operations. Lines 5- 11 are called at most $T\mathcal{N}$ times, and thus require $O(T\mathcal{N}\text{nnz}(A))$ arithmetic operations.

- Every time it is run, Line 25 requires computing $f(\theta)$ and $f(z)$, which can be done in 2 calls to the oracle for the value of $f$. It also requires computing $\|z-\theta\|_{\Phi(\theta)}^2$ and $\|\theta - z\|_{\Phi(z)}^2$. As

$$\|z - \theta\|_{\Phi(\theta)}^2 = (z - \theta)^\top\Phi(\theta)(z - \theta) = (z - \theta)^\top\alpha^{-1}\hat{A}^\top\hat{S}(\theta)^2\hat{A}(z - \theta),$$

this can be done in $O(\text{nnz}(A))$ arithmetic operations using standard matrix-vector multiplication. For the same reason, computing $\|\theta - z\|_{\Phi(z)}^2$ also takes $O(\text{nnz}(A))$ arithmetic operations. Line 25 is called at most $T$ times, and thus contributes $O(T(\text{nnz}(A) + T_f))$ arithmetic operations to the runtime.

- Each time they are run, Lines 18 and 21 require computing a Taylor series with at most $O(\mathcal{N}^2)$ terms, with each term requiring $O(\mathcal{N})$ multiplications. Lines 18 and 21 are called at most $T$ times and thus require $O(T\mathcal{N}^3)$ arithmetic operations.

- Line 2, which computes $\log\det(\hat{A}^\top\hat{A})$, can be accomplished in $O(md^{\omega-1})$ arithmetic operations using dense matrix multiplication. Since this step is only run once and $O(md^{\omega-1}) \leq O(T\mathcal{N}(\mathrm{nnz}(A) + d^2))$ arithmetic operations, it does not change the overall runtime by more than a constant factor.

*Runtime for all steps:* Thus, the runtime of Algorithm 1 is $O(T\mathcal{N}(T_f + \mathrm{nnz}(A) + d^2)) = O((md + dL^2R^2) \times \log^{2.01}(\frac{md + L^2R^2 + \log(w)}{\delta})\log(\frac{w}{\delta})) \times (T_f + \mathrm{nnz}(A) + d^2)$ arithmetic operations in the setting where $f$ is $L$-Lipschitz.

In the setting where $f$ is $\beta$-smooth, the number of arithmetic operations is $O(T\mathcal{N}(T_f + \mathrm{nnz}(A) + d^2)) = O((md + dL^2R^2) \times \log^{2.01}(\frac{md + \beta R^2 + \log(w)}{\delta})\log(\frac{w}{\delta})) \times (T_f + \mathrm{nnz}(A) + d^2)$.

**Bounding the total variation distance:** Plugging Lemma B.2 into Lemma B.3, we get that for every $j \in \{1, \ldots, \mathcal{N}\}$,

$$\mathbb{P}\left(|Y_j - \mathbb{E}[Y_j]| \leq \frac{1}{100}\right) \leq 1 - \gamma. \tag{21}$$

By Proposition A.7 we have that,

$$\begin{aligned}
\mathbb{E}[Y_j] &= \mathbb{E}[v^\top(t\hat{A}^\top\hat{S}(\theta)^{-2}\hat{S}(z)\hat{A} + (1-t)\hat{A}^\top\hat{A})^{-1}(\hat{A}^\top\hat{S}(\theta)^{-2}\hat{S}(z)\hat{A} - \hat{A}^\top\hat{A})v] \\
&\quad + \log\det(\hat{A}^\top\hat{A}) \\
&= \int_0^1 \mathrm{tr}\left(\tau\hat{A}^\top\hat{S}(\theta)^{-2}\hat{S}(z)\hat{A} + (1-\tau)\hat{A}^\top\hat{A})^{-1}(\hat{A}^\top\hat{S}(\theta)^{-2}\hat{S}(z)\hat{A} - \hat{A}^\top\hat{A}\right)\mathrm{d}\tau \\
&\quad + \log\det(\hat{A}^\top\hat{A}) \\
&\overset{\text{Prop. } A.7}{=} \log\det(\hat{A}^\top\hat{S}(\theta)^{-2}\hat{S}(z)\hat{A}) \\
&= \log\det(\Phi(z)) - \log\det(\Phi(\theta)),
\end{aligned} \tag{22}$$

where the second equality holds since Algorithm 1 samples the random vector $v \sim N(0, I_d)$ from the standard Gaussian distribution and the random variable $t \sim \mathrm{unif}([0,1])$ from the uniform distribution on $[0,1]$.

Moreover, by Lemma A.3 we have that

$$\begin{aligned}
\mathbb{P}\left(\mathbb{E}[Y_j] \geq \frac{9}{10}\right) &\overset{\text{Eq. }(22)}{=} \mathbb{P}\left(\log\det(\Phi(z)) - \log\det(\Phi(\theta)) \geq \frac{9}{10}\right) \\
&\overset{\text{Lemma }A.3}{\geq} 1 - \frac{1}{100}\gamma.
\end{aligned}$$

Next, we show that

$$\mathbb{P}(0 \leq X \leq 1) \geq 1 - 2\mathcal{N}\gamma. \tag{23}$$

and that

$$\left|\mathbb{E}[X] - \frac{\frac{\det(\Phi(z))}{\det(\Phi(\theta))}}{\frac{\det(\Phi(\theta))}{\det(\Phi(z))} + \frac{\det(\Phi(z))}{\det(\Phi(\theta))}}\right| \leq \gamma. \tag{24}$$

We first show (23) and (24) when $\frac{1}{4} \leq Y_1 \leq 2\log(\gamma)$. In this case we have by (21) that with probability at least $1 - \gamma$,

$$\log\det(\Phi(z)) - \log\det(\Phi(\theta) = \mathbb{E}[Y_1] \in \left[\frac{1}{4} - \frac{1}{100}, \ 2\log\frac{1}{\gamma}\right). \tag{25}$$

To show (23), we first note that, by (21) we have that,

$$\mathbb{P}\left(\frac{1}{5} \leq Y_j \leq 3\log\left(\frac{1}{\gamma}\right), \ \forall j \leq \ell \leq \mathcal{N}\right) \geq 1 - \mathcal{N}\gamma. \tag{26}$$

Thus, with probability at least $1 - \mathcal{N}\gamma$,

$$
\begin{aligned}
X &= 1 + \frac{1}{2} \sum_{k=1}^{2\mathcal{N}-1} (-1)^k \sum_{\ell=0}^{\mathcal{N}} \frac{1}{\ell!} \prod_{j=1}^{\ell} (-2kY_j) \\
&\leq 1 + \frac{1}{2} \sum_{k=1}^{2\mathcal{N}-1} (-1)^k \sum_{\ell=0}^{\mathcal{N}} \frac{1}{\ell!} \prod_{j=1}^{\ell} (-2k \sup_{1 \leq j \leq \mathcal{N}} Y_j) \\
&\leq 1 + \frac{1}{2} \sum_{k=1}^{2\mathcal{N}-1} (-1)^k \exp(-2k \sup_{1 \leq j \leq \mathcal{N}} Y_j) \\
&\leq 1 + \frac{1}{2} \sum_{k=1}^{2\mathcal{N}-1} (-1)^k \exp(-2k \sup_{1 \leq j \leq \mathcal{N}} Y_j) \\
&\leq \frac{1}{1 + \exp(-\sup_{1 \leq j \leq \mathcal{N}} Y_j)} \\
&\leq 1, \tag{27}
\end{aligned}
$$

where the fourth inequality holds since

$$
1 + \frac{1}{2} \sum_{k=1}^{2\mathcal{N}-1} (-1)^k \exp(-2kt) \leq 1 + \frac{1}{2} \sum_{k=1}^{\infty} (-1)^k \exp(-2kt) \qquad \forall t > 0,
$$

and since the infinite series $1 + \frac{1}{2} \sum_{k=1}^{\infty} (-1)^k \exp(-2kt) = \frac{1}{2} \tanh(\frac{1}{2}t) + 1 = \frac{1}{1+e^{-t}}$ has interval of convergence $t \in (0, \infty]$.

Next, we show that with probability at least $1 - \mathcal{N}\gamma$,

$$
\begin{aligned}
X &= 1 + \frac{1}{2} \sum_{k=1}^{2\mathcal{N}-1} (-1)^k \sum_{\ell=0}^{\mathcal{N}} \frac{1}{\ell!} \prod_{j=1}^{\ell} (-2kY_j) \\
&\geq 1 + \frac{1}{2} \sum_{k=1}^{2\mathcal{N}-1} (-1)^k \sum_{\ell=0}^{\mathcal{N}} \frac{1}{\ell!} \prod_{j=1}^{\ell} (-2k \inf_{1 \leq j \leq \mathcal{N}} Y_j) \\
&\geq 1 + \frac{1}{2} \sum_{k=1}^{2\mathcal{N}-1} (-1)^k \exp(-2k \inf_{1 \leq j \leq \mathcal{N}} Y_j) \\
&= 1 - \exp(-4\mathcal{N} \inf_{j \leq \ell \leq \mathcal{N}} Y_j) + \frac{1}{2} \sum_{k=1}^{2\mathcal{N}} (-1)^k \exp(-2k \inf_{1 \leq j \leq \mathcal{N}} Y_j) \\
&\geq \frac{1}{1 + \exp(-\inf_{1 \leq j \leq \mathcal{N}} Y_j)} \\
&\geq \frac{1}{2} - \exp(-4\mathcal{N} \inf_{1 \leq j \leq \mathcal{N}} Y_j) \\
&> 0, \tag{28}
\end{aligned}
$$

where the last inequality holds since $\inf_{1 \leq j \leq \mathcal{N}} Y_j \geq \frac{1}{8}$ with probability at least $1 - \mathcal{N}\gamma$ by (26). Thus, (27) and (28) together imply (23).

To show (24), we first note that, since $Y_1, Y_2, \ldots, Y_{\mathcal{N}}$ are i.i.d. random variables, with probability at least $1 - \mathcal{N}\gamma$,

$$
\begin{aligned}
\mathbb{E}[X] &= 1 + \frac{1}{2} \sum_{k=1}^{2\mathcal{N}-1} \sum_{\ell=0}^{\mathcal{N}} (-1)^k \frac{1}{\ell!} \prod_{j=1}^{\ell} (-2k\mathbb{E}[Y_j]) \\
&= 1 + \frac{1}{2} \sum_{k=1}^{2\mathcal{N}-1} (-1)^k \sum_{\ell=0}^{\mathcal{N}} \frac{1}{\ell!} \prod_{j=1}^{\ell} (-2k(\log\det(\Phi(z)) - \log\det(\Phi(\theta)))) \\
&= 1 + \frac{1}{2} \sum_{k=1}^{2\mathcal{N}-1} (-1)^k \sum_{\ell=0}^{\mathcal{N}} \frac{1}{\ell!} (-2k(\log\det(\Phi(z)) - \log\det(\Phi(\theta))))^{\ell} \\
&= \frac{1}{1 + \exp(-\log\det(\Phi(z)) - \log\det(\Phi(\theta)))} \\
&\quad - \frac{1}{2} \sum_{k=1}^{2\mathcal{N}-1} (-1)^k \sum_{\ell=\mathcal{N}+1}^{\infty} \frac{1}{\ell!} (-2k(\log\det(\Phi(z)) - \log\det(\Phi(\theta)))^{\ell} \\
&\quad - \frac{1}{2} \sum_{k=2\mathcal{N}}^{\infty} (-1)^k \sum_{\ell=0}^{\infty} \frac{1}{\ell!} (-2k(\log\det(\Phi(z)) - \log\det(\Phi(\theta)))^{\ell}, \\
&= \frac{1}{1 + \exp(-\log\det(\Phi(z)) - \log\det(\Phi(\theta)))} \\
&\quad - \frac{1}{2} \sum_{k=1}^{2\mathcal{N}-1} (-1)^k \sum_{\ell=\mathcal{N}+1}^{\infty} \frac{1}{\ell!} (-2k(\log\det(\Phi(z)) - \log\det(\Phi(\theta))))^{\ell} \\
&\quad - \frac{1}{2} \sum_{k=2\mathcal{N}}^{\infty} (-1)^k \exp(-2k(\log\det(\Phi(z)) - \log\det(\Phi(\theta)))),
\end{aligned}
$$

where the fourth equality holds since the infinite series $1 + \frac{1}{2}\sum_{k=1}^{\infty}(-1)^k \exp(-2kt) = \frac{1}{1+e^{-t}}$ has interval of convergence $t \in (0, \infty]$, and $\log\det(\Phi(z)) - \log\det(\Phi(\theta) \in \left[\frac{1}{4} - \frac{1}{100},\ 2\log(\frac{1}{\gamma})\right)$ by (25).

Thus,

$$
\begin{aligned}
&\left| \mathbb{E}[X] - \frac{\frac{\det(\Phi(z))}{\det(\Phi(\theta))}}{\frac{\det(\Phi(\theta))}{\det(\Phi(z))} + \frac{\det(\Phi(z))}{\det(\Phi(\theta))}} \right| \\
&= \left| \mathbb{E}[X] - \frac{1}{1 + \exp(-\log\det(\Phi(z)) - \log\det(\Phi(\theta)))} \right| \\
&= \left| \frac{1}{2} \sum_{k=1}^{\infty} (-1)^k \sum_{\ell=\mathcal{N}+1}^{\infty} \frac{1}{\ell!} (-2k(\log\det(\Phi(z)) - \log\det(\Phi(\theta))))^{\ell} \right. \\
&\quad \left. + \frac{1}{2} \sum_{k=2\mathcal{N}}^{\infty} (-1)^k \exp(-2k(\log\det(\Phi(z)) - \log\det(\Phi(\theta)))), \right| \\
&\leq \gamma,
\end{aligned}
$$

since $\mathcal{N} \geq 10\log(\frac{1}{\gamma})$ and $\log\det(\Phi(z)) - \log\det(\Phi(\theta) \in \left[\frac{1}{4} - \frac{1}{100},\ 2\log(\frac{1}{\gamma})\right)$. This proves (24).

Thus, we have shown (24) and (23) in the setting when $\frac{1}{4} \leq Y_1 \leq 2\log(\gamma)$.

The proof of (24) and (23) when $Y_1 > \frac{1}{4}$ is identical, except that we use the Taylor series expansion for $\frac{1}{1+e^{-t}} = \frac{1}{2}\tanh(\frac{1}{2}t) + 1$ about 0, $\frac{1}{1+e^{-t}} = \sum_{\ell=0}^{\infty} c_\ell t^\ell$ which has interval of convergence $t \in (-1, 1)$, in place of the infinite series $1 + \frac{1}{2}\sum_{k=1}^{\infty}(-1)^k \exp(-2kt) = \frac{1}{2}\tanh(\frac{1}{2}t) + 1 = \frac{1}{1+e^{-t}}$ which has interval of convergence $t \in (0, \infty]$.

The proof for (24) and (23) in the case when $Y_1 > 2\log(\frac{1}{\gamma})$ is trivial, as in this case the algorithm sets $X = 1$ and $|1 - \frac{1}{1+e^{-t}}| \le \gamma$ for $t > 2\log(\gamma)$.

Thus, (24) and (23) together imply that

$$\left| \mathbb{E}[\min(\max(X,0),1)] - \frac{\frac{\det(\Phi(z))}{\det(\Phi(\theta))}}{\frac{\det(\Phi(\theta))}{\det(\Phi(z))} + \frac{\det(\Phi(z))}{\det(\Phi(\theta))}} \right| \le 3\mathcal{N}\gamma. \tag{29}$$

Let $q_i = \mathbb{E}[\frac{1}{2}\min(\max(X,0),1)] \times \min\left(\frac{e^{-f(z)}}{e^{-f(\theta)}} \times e^{\|z-\theta\|^2_{\Phi(\theta)} - \|\theta-z\|^2_{\Phi(z)}}, 1\right)$ denote the probability that the proposed step $z$ is accepted at any iteration $i$ of Algorithm 1. Then for every $i \in \{1, \ldots, T\}$, we have by (29) that

$$\left| q_i - \frac{1}{2}\frac{\frac{\det(\Phi(z))}{\det(\Phi(\theta))}}{\frac{\det(\Phi(\theta))}{\det(\Phi(z))} + \frac{\det(\Phi(z))}{\det(\Phi(\theta))}} \times \min\left(\frac{e^{-f(z)}}{e^{-f(\theta)}} \times e^{\|z-\theta\|^2_{\Phi(\theta)} - \|\theta-z\|^2_{\Phi(z)}}, 1\right) \right| \le 3\mathcal{N}\gamma \tag{30}$$

Let $\mathcal{X}_0, \mathcal{X}_1, \ldots$ be the Markov chain where at the beginning of each iteration $i+1$ of Algorithm 1, $\mathcal{X}_i = \theta$. Moreover, let $\mathcal{Y}_0, \mathcal{Y}_1, \ldots$ be the Markov chain defined in Algorithm 2. We define a probabilistic coupling between these two Markov chains. First, define a coupling between the random vector $\xi \equiv \xi_i$ sampled at every iteration $i+1$ of Algorithm 1 and the random vector $\zeta_i$ sampled at every iteration $i$ of Algorithm 2, such that $\xi_i = \zeta_i$ for all $i \ge 0$. Then, by (30), there exists a coupling between the Markov chains $\mathcal{X}_0, \mathcal{X}_1, \ldots$ and $\mathcal{Y}_0, \mathcal{Y}_1, \ldots$ such that with probability at least $1 - 3T\mathcal{N}\gamma$,

$$\mathcal{X}_i = \mathcal{Y}_i \qquad \forall i \in \{0, 1, \ldots, T\}. \tag{31}$$

Thus, letting $\mu$ denote the distribution of the output $\mathcal{X}_T$ of Algorithm 1, and letting $\nu_T$ denote the distribution of $\mathcal{Y}_T$, we have by (31) that

$$\|\mu - \nu_T\|_{\mathrm{TV}} \le 3T\mathcal{N}\gamma \le \frac{1}{2}\delta. \tag{32}$$

Moreover, by Lemma A.5, we have that the distribution $\nu_T$ of $\mathcal{Y}_T$ satisfies

$$\|\nu_T - \pi\|_{\mathrm{TV}} \le \frac{1}{2}\delta. \tag{33}$$

Thus, we have that the distribution $\mu$ of the output of Algorithm 1 satisfies

$$\|\mu - \pi\|_{\mathrm{TV}} \le \|\mu - \nu_T\|_{\mathrm{TV}} + \|\nu_T - \pi\|_{\mathrm{TV}} \le \delta.$$

■

# C  ADDITIONAL COMPARISONS OF APPROACHES TO DETERMINANT COMPUTATION

In this section we explain the key differences between our randomized estimator for the determinant term, and the estimator of Laddha et al. (2020), and why these differences are necessary.

**Approach of Laddha et al. (2020):** In Laddha et al. (2020), the determinant ratio is first estimated by plugging in independent random estimates for the log-determinant with mean $\log \det H(\theta) - \log \det H(z)$ into the Taylor expansion for the exponential function $e^t$. This gives an estimate $V$ with mean $\mathbb{E}[V] = \frac{\det(H(\theta))}{\det(H(z))}$ (their Lemma 4.3). Roughly speaking, the authors then suggest plugging this estimate $V$ (and another estimate $\hat{V}$ with mean $\frac{\det H(z)}{\det(H(\theta))}$) into a smooth function $g(t,s)$, to compute an estimate for their Metropolis acceptance rule $\frac{p(z \to \theta)}{p(z \to \theta) + p(\theta \to z)} = g(\frac{\det(H(\theta))}{\det(H(z))}, \frac{\det(H(z))}{\det(H(\theta))})$. Each time their Markov chain proposes an update $\theta \to z$, it is accepted with probability $g(V, \hat{V})$.

However, the randomized estimator $g(V, \hat{V})$ is not an unbiased estimator for the Metropolis rule $g(\frac{\det(H(\theta))}{\det(H(z))}, \frac{\det(H(z))}{\det(H(\theta))})$, even though $\mathbb{E}[V] = \frac{\det(H(\theta))}{\det(H(z))}$ and $\mathbb{E}[\hat{V}] = \frac{\det(H(z))}{\det(H(\theta))}$. This is because, in

general, $\mathbb{E}[g(V, \hat{V})] \neq g(\mathbb{E}[V], \mathbb{E}[\hat{V}])$ since $g(V, \hat{V})$ is not a linear function in $V$ or $\hat{V}$. If one utilizes this randomized estimator in the acceptance rule of the Dikin walk Markov chain, it can cause the Markov chain to generate samples from a distribution not equal to the (uniform) target distribution. This problem does not seem to be easily fixable, and requires a different approach.

**Our approach to estimating the determinantal term:** To obtain an unbiased estimator for the determinantal term, our algorithm bypasses the use of the Taylor expansion for the exponential function $e^t$ used in Laddha et al. (2020), and instead uses two different infinite series expansions for the sigmoid function (a Taylor series expansion $\mathrm{sigmoid}(t) = \sum_{i=0}^{\infty} c_i t^i$ centered at 0 with region of convergence $(\pi, \pi)$, and a series expansion "at $+\infty$" with region of convergence $(0, +\infty)$ which is polynomial in $e^{-t}$) to compute an unbiased estimate with mean equal to a (different) Metropolis acceptance rule. The Metropolis rule in our algorithm is proportional to $\mathrm{sigmoid}\left(\frac{1}{2}\log\det(\Phi(z)) - \frac{1}{2}\log\det(\Phi(\theta))\right)$. To generate a randomized estimator for this acceptance rule, we compute independent samples $Y_1, \ldots, Y_n$ with mean $\log\det(\Phi(\theta)) - \log\det(\Phi(z))$, and plug these samples into one of the two series expansions, e.g. $\sum_{i=0}^{N} c_i Y_1 \cdot \ldots \cdot Y_i$, for some small number of terms $N$. To select which series expansion to use, our algorithm chooses a series expansion whose region of convergence contains a ball of radius $\frac{1}{8}$ centered at $Y_1$.

Since $Y_1, \ldots, Y_N$ are independent, roughly speaking, the mean of our estimator is proportional to

$$\mathbb{E}\left[\sum_{i=0}^{N} c_i Y_1 \cdot \ldots \cdot Y_i\right] = \sum_{i=0}^{N} c_i \mathbb{E}[Y_1] \cdot \ldots \cdot \mathbb{E}[Y_i] = \sum_{i=0}^{N} c_i (\log\det(\Phi(\theta)) - \log\det(\Phi(z)))^i,$$

(or to a similar quantity if the other series expansion is selected). We show that, w.h.p., the choice of series expansion selected by our algorithm converges exponentially fast in $N$ when $Y_1, \ldots, Y_N$ are plugged into this expansion. This implies that, if we choose roughly $N = \log\frac{1}{\gamma}$, our estimator is a (nearly) unbiased estimator with mean within any desired error $\gamma > 0$ of the correct value $\mathrm{sigmoid}\left(\frac{1}{2}\log\det(\Phi(z)) - \frac{1}{2}\log\det(\Phi(\theta))\right)$. We then show that this error $\gamma$ is sufficient to guarantee that our Markov chain samples within total variation error $O(\delta)$ from the correct target distribution, if one chooses $\gamma = O(\mathrm{poly}(\delta, 1/d, 1/L))$.

