# OpenReview forum: "Faster Sampling from Log-Concave Densities over Polytopes via Efficient Linear Solvers"
_ICLR.cc/2024/Conference — ICLR 2024 poster_

### Official Review · Reviewer_mHXa · 2023-10-13

**Soundness:** 3 good
**Presentation:** 3 good
**Contribution:** 2 fair
**Rating:** 3
**Confidence:** 5

**Summary:**

This paper considers using a soft-threshold Dikin walk to sample from a polytope contained in a box of radius $R$. The sampling distribution is log-concave and specified by a Lipschitz/smooth function $f$. By using a pipeline introduced in [LLV20], they show how to speed up the algorithm introduced in [MV23] so that each iteration, instead of computing the Hessian of log-barrier using fast matrix multiplication, one can resort to the inverse maintenance data structure due to [LS15]. To this end, they achieve a similar per iteration cost as [LLV20], namely $\mathrm{nnz}(A)+d^2$.

**Strengths:**

This paper improves the per iteration cost of [MV23] from $md^{\omega-1}$ to $\mathrm{nnz}(A)+d^2$. In most regimes where $m\geq d$ and $A$ is relatively sparse, this is a strict upgrade from prior state-of-the-art.

**Weaknesses:**

I'm very dubious of the novelty of this paper. What's the major difference between the algorithm in this paper and [LLV20]? While this paper only mentions [LLV20] sparingly, the algorithmic framework is almost identical. Specifically, [LLV20] shows that for sampling uniformly over a polytope with log-barrier, one can use the [LS15] inverse maintenance to compute a new sampling point. The determinant ratio term can be estimated via an unbiased estimator, which can further be estimated using Taylor expansion together with terms that can be quickly computed using the inverse maintenance data structure. The only difference is that this paper also needs to handle the regularization term, but it is neither surprising nor novel the machinery of [LLV20] also works here.

It is worth noting that [LLV20] does not provide any proof on why using an approximate solver and samples, the algorithm still converges. This paper provides a very simple argument to show it indeed works.

Overall, I think this paper should provide comprehensive comparison with algorithm in [LLV20]. What's the difference here? What's new? Otherwise, it should acknowledge that the algorithm is largely derivative from [LLV20]. In its current writing, the authors acknowledge the fast linear solver part follows from [LLV20]. What about the determinant ratio part? What's the main difference between your approach for determinant and [LLV20]? What's the novelty of your algorithm?

**Questions:**

See weakness.

---

> ### Author Response · Authors · 2023-11-20
>
> Thank you for your valuable comments and suggestions.  We address your questions and concerns below, specifically, in relation  to [LLV20]. Sorry about the confusion-- while we discuss the differences in the two papers in the proof overview (Section 4), we would be happy to make things more explicit and add a separate section outlining these differences.
>
> In short, the key differences between the algorithm and techniques in our paper and those of [LLV20] are as follows:
>
> 1. We use a different randomized estimator for the determinantal term in the Metropolis acceptance probability. We need a different estimator because the randomized estimator for the determinantal term in the Metropolis acceptance rule of [LLV20] does not lead to a valid unbiased estimator for the correct acceptance probability (even for the special case $f=0$). We think this is an error in their paper, and it is not easily fixable.
> 2. Additionally, we handle the more general case where $f \neq 0$. This requires us to use a Markov chain based on an $\ell_2$-regularized barrier function of [MV23], and leads to additional challenges in bounding the change of the Hessian matrix of the barrier function at each step of our algorithm.
>
> We explain each of these differences in detail below. If you would like to see these details in a separate section in a revision, we would be happy to do so. We hope you will reconsider your opinion of our paper.  Please feel free to reach out if something is not clear, or if you have additional questions.

---

> > ### Comment · Reviewer_mHXa · 2023-11-20
> >
> > Thank you for your response. I agree with the first assessment, the [LLV20] acceptance rule likely has some issues even for uniform case. A fix for that is appreciated. However, this does not address my major concern of the paper --- from my point of view, this paper pretty straightforwardly combines the last section of [LLV20] and the soft-threshold Dikin walk in [MV23]. I strongly believe to justify the acceptance of this paper, authors should try to add more results in addition to the current $A+B$ result. For example, can you extend soft-threshold Dikin walk to other barriers for a better mixing? For more intricate barriers such as hybrid barrier or Lee-Sidford barrier, can you analogously design this [LLV20]-like fast solvers for them? At its current form, I will keep my score as is.
> >
> > P.S. (Not for this version), in the next revision, consider comparing with this paper: [KV23]. Authors show that it's possible to do log-concave sampling over polytopes using other barriers such as Lee-Sidford, get better mixing and without dependence on $R$. Moreover, they only require Lipschitz on the density instead of $f$. On the other hand, they do need $f$ to be $\alpha$-relatively strong convex. I think no polynomial dependence on $R$ is essential for Dikin walk, as this (used to be) a key advantage over hit-and-run. Can you extend the [LLV20] machinery to their Dikin walk?
> >
> > [KV23] Y. Kook and S. Vempala. Gaussian Cooling and Dikin Walks: The Interior-Point Method for Logconcave Sampling. 2023.

---

> > > ### Author Response · Authors · 2023-11-21
> > >
> > > Thank you for your prompt reply. We took your original review seriously and made an earnest attempt to address all your questions and concerns. In particular, you had asked us "What's the major difference between the algorithm in this paper and [LLV20]?" and “What's the main difference between your approach for determinant and [LLV20]? What's the novelty of your algorithm?”. We believe that we have addressed all the questions and concerns in your original review. We were in the middle of posting our responses to your original review when we noticed your reply to the first part of our response.
> > >
> > > We are pleased to see that you agree to the fact that the approach in [LLV20] had issues (even for the uniform case) and appreciate that our approach resolves these issues. However, we find it unfair that instead of raising the score, your response has raised the bar for accepting the paper by asking us to provide substantial additional results which are not mentioned in your original review.  Moreover, we are perplexed by the confidence of 5 which corresponds to "You are absolutely certain about your assessment. You are very familiar with the related work and checked the math/other details carefully," given that the original review overlooked important details in both [LLV20] and our paper.
> > >
> > > As for your comment in the recent reply which characterizes our paper as an "A+B result", we believe this is an innacurate representation of our work: Significant novel work was required to make the [LLV20] component work, extend the [MV23] results to improve the per-step complexity when $f \neq 0$  (which required bounding the change in the Hessian matrix of the $\ell_2$-regularized barrier function used in their Markov chain) and to combine these results with the fast linear solver of [LS15]. The synthesis of these results is non-trivial and implies the fastest runtime for important ML applications of sampling to training Bayesian models and differential privacy.
> > >
> > > As for the condition for accepting our paper in your recent response, which requires proving  results that reduce the number of Markov chain iterations, obtaining these results is a distinct and non-trivial direction, as mentioned in [MV23], even in the simpler setting where one implements the Markov chain with matrix multiplication-based linear solvers. The focus of our paper is on improving the per-step complexity of the Markov chain, which is motivated from practical problems. Therefore, we do not believe including these additional results would make sense for our paper.
> > >
> > > We would be happy to answer any further questions related to our paper. We hope you will consider our responses while formulating your opinion.

---

> > > > ### Comment · Reviewer_mHXa · 2023-11-21
> > > >
> > > > Thank you for your followup. I apologize to authors for my initial review on the part of determinant term. Now I do understand that the authors try to fix a potential issue in [LLV20] without explicitly pointing out the possible errors in that paper. I chose confidence 5 in my initial review because I've tried to work on followups on both [LLV20] and [MV23] and have thoroughly studied the techniques and proofs from both of the papers. I acknowledge I've overlooked the determinant fix part in this paper and would like to apologize to the authors.
> > > >
> > > > However, I stand by my initial assessment that this paper lacks novelty. In my opinion, the main virtue of this paper is the fix and possibly a clearer explanation of the inverse maintenance algorithm in the last section of [LLV20]. However, I do feel this paper a bit too derivative from [MV23]. As authors mentioned in how they prove the data structure in turn works for their algorithm, they mainly use an argument from [MV23], saying that the regularized Hessian is actually the Hessian of log-barrier of *some* polytopes, therefore operating on the regularized Hessian is essentially ``change of variable'' on the Hessian of log-barrier. Once one realizes this reduction, then the framework and analysis from [LLV20] can be adapted, provided a few extra relatively straightforward lemmas on the regularized Hessian. In my opinion, the work [MV23] makes a large Delta in the sense that it bridges the gaps between uniform and log-concave sampling over polytopes using log-barrier Dikin walk, even though it has some polynomial dependence on $R$. This paper on the other hand, is a much smaller Delta from [MV23]: it fixes an issue in the algorithm of [LLV20] and deploys it to the [MV23] walk. Most of the analyses are also consequence of [MV23]. This is the major reason I asked for more results from my prior response, as at its current form, I feel the contribution of this paper is not significant enough to warrant an accept.
> > > >
> > > > I hope this clarifies my prior response and my rating. Again, thank you for your followup.

---

> > > > > ### Author Response · Authors · 2023-11-22
> > > > >
> > > > > Thank you for your followup. We feel like the goal-post for accepting our paper has been changing.
> > > > >
> > > > > In your original review your main concern was how our paper differs from [LLV20], and we explained this in detail in our reponse.  You seem to agree that [LLV20] had important issues which we resolved and which you missed.
> > > > >
> > > > > In your next reply, you referred to our paper as an "A+B result" and suggested a new condition for acceptance involving improving the Dikin walk in an orthogonal direction. We explained in our subsequent response how our result is not an "A+B result", and pointed out that the open problem you suggested is actually hard even in the [MV23] setting and would constitute a different paper.
> > > > >
> > > > > *In reality, our paper fixes a major issue in [LLV20] which was published in STOC 2020, and seems to have solved an open problem in [MV23] which appears in NeurIPS 2023 (see Section 4 [here](https://openreview.net/pdf?id=BA7NHAzbpO)).* Our paper does so via a non-trivial synthesis and augmentation of results from [LS15], [LLV20], and [MV23]. It also implies a significant speed-up for the runtime of several practical problems arising in ML. We remain uncertain how this qualifies as a clear reject from ICLR as per your score. It is also hard to understand the confidence rating, given that several important details have been missed.
> > > > >
> > > > > We would be happy to clarify these connections further in the final version of the paper and hope you will reconsider your score. We would also be happy to continue this thread if you have further questions about our paper.

---

> > > > > > ### Comment · Reviewer_mHXa · 2023-11-22
> > > > > >
> > > > > > Thanks authors for the followup. I think I've made my points quite clear in the prior response. The major contribution of this paper is a (potential) fix of the determinant estimation part of [LLV20]. Beyond that, I don't see explicitly where do you need *nontrivial* synthesis between [LLV20] and [MV23]. The algorithm is essentially identical to that of [LLV20] with a slightly different determinant estimation (it should be noted the fix also follows the general strategy of [LLV20] with a different acceptance rule and series, but the way it applies inverse maintenance is on a very similar term).
> > > > > >
> > > > > > I've never raised the bar for acceptance and have made myself very clear in the prior response. This paper makes a small Delta from [MV23], thus I feel the contributions are limited.
> > > > > >
> > > > > > Finally, as authors repeatedly mention they fix a major issue of [LLV20] (which I mostly agree, but this does not affect the main result of [LLV20] as theirs is identifying a sufficient condition for Dikin walk from uniform distribution, so that one can finally obtain a $d^2$ mixing time using Lee-Sidford barrier, improving the $d^{2.5}$ bound from [CDWY18]), then they should contain a section discussing the issues in [LLV20] and how they manage to fix it. This will improve the paper's presentation and signify the main contribution of this paper, albeit a different acceptance rule and series for logdet from [LLV20].

---

> > > > > > > ### Author Response · Authors · 2023-11-23
> > > > > > >
> > > > > > > Thank you for your response. We will add a section directly comparing our work to [LLV20] in the final version.
> > > > > > >
> > > > > > > We are grateful for the time you have dedicated to the review process.

---

> > ### Author Response · Authors · 2023-11-20
> >
> > **1. Differences between our randomized estimator for the determinant term, and the estimator of [LLV20]:**
> >
> > ***Approach of [LLV20]:*** In [LLV20], the determinant ratio is first estimated by plugging in independent random estimates for the log-determinant with mean $\mathrm{logdet} (H(\theta))- \mathrm{logdet}(H(z))$  into the Taylor expansion for the exponential function $e^t$. This gives an estimate $V$ with mean $E[V] = \frac{\mathrm{det} (H(\theta))}{\mathrm{det}(H(z))}$ (their Lemma 4.3).
> > Roughly speaking, the authors then suggest plugging this estimate $V$ (and another estimate $\hat{V}$ with mean $\frac{\mathrm{det} (H(z))}{\mathrm{det}(H(\theta))}$) into a smooth function $g(t,s)$, to compute an estimate for their Metropolis acceptance rule $\frac{p(z\rightarrow \theta)}{ p(z\rightarrow \theta) + p(\theta\rightarrow z)} = g(\frac{\mathrm{det} (H(\theta))}{\mathrm{det}(H(z))}, \frac{\mathrm{det} (H(z))}{\mathrm{det}(H(\theta))})$.  Each time their Markov chain proposes an update $\theta \rightarrow z$, it is accepted with probability $g(V, \hat{V})$.
> >
> > However, the randomized estimator $g(V, \hat{V})$ used in [LLV20] is not an unbiased estimator for their Metropolis rule $g(\frac{\mathrm{det} (H(\theta))}{\mathrm{det}(H(z))}, \frac{\mathrm{det} (H(z))}{\mathrm{det}(H(\theta))})$,  even though $E[V] = \frac{\mathrm{det}(H(\theta))}{\mathrm{det}(H(z))}$ and $E[\hat{V}] = \frac{\mathrm{det} (H(z))}{\mathrm{det}(H(\theta)))}$.  This is because, in general, $E[g(V, \hat{V})] \neq g(E[V], E[\hat{V}])$ since $g(V, \hat{V})$ is not a linear function in $V$ or $\hat{V}$.
> > (One can construct a counter-example where the above equality does not hold.) This causes their Markov chain to generate samples from a distribution not equal to the (uniform) target distribution considered in their paper. This problem does not seem to be easily fixable, and requires a different approach.
> >
> > ***Our approach to estimating the determinantal term:*** To obtain an *unbiased* estimator for the determinantal term, our algorithm bypasses the use of the Taylor expansion for the exponential function $e^t$ used in [LLV20],  and instead uses two different infinite series expansions for the sigmoid function (a Taylor series expansion $\mathrm{sigmoid}(t) = \sum_{i=0}^\infty c_i t^i$  centered at $0$ with region of convergence $(-\pi, \pi)$, and a series expansion "at $+ \infty$" with region of convergence $(0, +\infty)$, which is a polynomial in $e^{-t}$) to compute an unbiased estimate with mean equal to a (different) Metropolis acceptance rule. Our Metropolis rule is proportional to $\mathrm{sigmoid}\left(\frac{1}{2}\log \mathrm{det}(\Phi(z)) - \frac{1}{2}\log \mathrm{det}(\Phi(\theta))\right).$  To generate a randomized estimator for this acceptance rule, we compute independent samples $Y_1,\cdots, Y_n$  with mean $\log \mathrm{det} \Phi(\theta) - \log \mathrm{det} \Phi(z)$, and plug these samples into one of the two series expansions, e.g. $\sum_{i=0}^N c_i Y_1…Y_i$, for some small number of terms $N$.  To select which series expansion to use, our algorithm chooses a series expansion whose region of convergence contains a ball of radius 1/8 centered at $Y_1$.
> >
> > Since $Y_1,…,Y_N$ are independent,  roughly speaking, the mean of our estimator is proportional to $E[\sum_{i=0}^N c_i Y_1 \cdots Y_i] =  \sum_{i=0}^N c_i E[Y_1]\cdots E[Y_i] =  \sum_{i=0}^N c_i (\log \mathrm{det} \Phi(\theta) - \log \mathrm{det} \Phi(z))^i$ (or to a simmilar quantity if the other series expansion is selected).  We show that, w.h.p., the choice of series expansion selected by our algorithm converges exponentially fast in $N$ when $Y_1,…,Y_N$ are plugged into this expansion (see our proof overview).  This implies that, if we choose roughly $N = \log\frac{1}{\gamma}$, our estimator is a (nearly) unbiased estimator with mean within any desired error $\gamma>0$ of the correct value $\mathrm{sigmoid}\left(\frac{1}{2}\log \mathrm{det}(\Phi(z)) - \frac{1}{2}\log \mathrm{det}(\Phi(\theta))\right)$.  We then show that this error $\gamma$ is sufficient to guarantee that our Markov chain samples within total variation error $O(\delta)$ from the correct target distribution, if one chooses $\gamma= O(\mathrm{poly}(\delta, 1/d, 1/L))$.

---

> > > ### Author Response · Authors · 2023-11-20
> > >
> > > **2. Differences in algorithm and proof arising from the more general target distribution our algorithm samples from:**
> > >
> > > In contrast to [LLV20], which considers only the special case when the target distribution $\pi$ is uniform on a polytope $K$, we handle the more general case where $\pi \propto e^{-f}$ may be any log-concave function with Lipschitz or smooth $f$  constrained to $K$.  To handle the general case, we give an efficient implementation of the "soft-threshold" Dikin walk of [MV23], whose steps are determined by an $\ell_2$-regularized barrier function.
> > >
> > > While the Hessian of the log-barrier function (as well as another barrier function) was previously shown in [LLV20] to change slowly w.r.t. the Frobenius norm, we cannot apply their bound to our algorithm.  This is because the regularized log-barrier Hessian, $\Phi$, used in the soft-threshold Dikin walk is not the Hessian of a log-barrier function for any set of equations defining the polytope K.
> > >
> > > To overcome this difficulty, we use the fact from [MV23] that the Hessian of the regularized barrier function is the limit of an infinite sequence of matrix-valued functions $H_j(\theta)$, $j=1,2,\cdots$, each of which *is* a Hessian of a (different) log-barrier function for the same polytope $K$.  This allows us to show that the regularized log-barrier function changes slowly with respect to the Frobenius norm at each step of the soft-threshold Dikin walk. In particular, we show that a rescaling $\Psi$ of the Hessian of the regularized barrier function satisfies the following inequality
> > > $$||(\Psi(z) - \Psi(\theta))^{-1} \Psi^{-1}(\theta) ||_F \leq O(1)$$
> > > with high probability. We then use this fact to show that a linear solver for $\Phi(\theta)$ can be maintained in time $\mathrm{nnz}(A) + d^2$ at each step $\theta$ of the Dikin walk using the efficient inverse maintenance algorithm of [LS15].

---

### Official Review · Reviewer_CJYs · 2023-11-01

**Soundness:** 3 good
**Presentation:** 3 good
**Contribution:** 4 excellent
**Rating:** 8
**Confidence:** 3

**Summary:**

The paper presents the current fastest algorithm for sampling from a log-concave distribution over a given polytope. In particular, they implement a more optimized version of the algorithm proposed by Mangoubi and Vishnoi [1], which is to appear in NeurIPS 2023, the arXiv version of which has been around for more than a year at this point. It is important to understand what [1] does, as the current paper heavily builds on it...

The algorithms considered in these papers are Metropolis-Hastings algorithms, which are a subclass of Markov chain Monte Carlo (MCMC) methods. For each step of the Markov chain, [1] needs time $O(md^{\omega-1})$ as it involves some matrix inversion and determinant computation steps, which can all be done in $O(md^{\omega-1})$ time by using algorithms for these problems which work for arbitrary matrices. The main claim in the current paper is that instead of recomputing these matrix operations from scratch in every step of the Markov chain, we can use the information from the previous step to speed up the computations of the current step.

References:

1: Sampling from Log-Concave Distributions over Polytopes via a Soft-Threshold Dikin Walk, Mangoubi, Oren and Vishnoi, Nisheeth K, arXiv preprint arXiv:2206.09384. To appear in NeurIPS 2023: https://neurips.cc/virtual/2023/poster/72502

**Strengths:**

Significance: The problem of sampling from a log-concave distribution over a polytope is interesting and also has several ML applications.

Originality: Clearly, the paper is an advance over the state-of-the-art. However, I am not super sure about how original this is in terms of technical contributions...

Clarity: Paper is clear but a bit dense and slightly hard to read, especially if you are not already familiar with Mangoubi and Vishnoi [1]. I re-read this paper after reading [1] and it was much clearer the second time. It might have to do with space-constraints, as the arXiv version of [1] obviously has more space so can go over a lot of things in more detail...

Quality: Overall seems to be of good quality. But I have not checked the correctness of the technical details. As it builds quite heavily on [1] and also a bunch of other results which I am not familiar with, I don't have the expertise to ascertain the correctness of the paper. But at least on the surface, it does seem like there are no major technical issues.

**Weaknesses:**

I don't have too much to point here, except what I already wrote about clarity before. The authors should try to make the paper more accessible to people who might not have read [1]. I know this is a generic remark but honestly I have no clear idea how to improve the paper in terms of clarity... There were a bunch of typos here and there, some of which are:

In Theorem 2.1, 3) convex function $f:K \mapsto \mathbb{R}^d$. Here $f$ should be $K \mapsto \mathbb{R}$. I saw this repeated in a bunch of other places. I think it's a typo but since I saw it repeatedly, both in this paper, and also in [1], I tried to dig around to see if it makes sense for the codomain of a convex function to be $\mathbb{R}^d$ instead of $\mathbb{R}$ but in that case, $e^{-f}$ would not be real, but it has to be real for it to be a probability density function?

Page 5, line 3: ball "or" radius -> ball of radius, again: $f: K \mapsto \mathbb{R}^d$

Page 7, two lines below inequality (3): “arithemtic”

**Questions:**

No additional questions

---

> ### Author Response · Authors · 2023-11-20
>
> Thank you for your valuable comments and suggestions. We are glad you find our contributions to be an advance over the state-of-the-art, and thank you for supporting our paper. We address your concerns below.
>
> >The authors should try to make the paper more accessible to people who might not have read [1].
>
> Thank you for your suggestion. We will add a section in the appendix providing additional background on the soft-threshold Dikin walk Markov chain of [1].
>
>
> >convex function $f :K \rightarrow \mathbb{R}^d$.  Here $f$ should be $K \rightarrow \mathbb{R}$.
>
> Sorry for the typo. It should read $f :K \rightarrow \mathbb{R}$, where $K\subset \mathbb{R}^d$ is a convex polytope.
>
>
> >Other typos:
>
> Thank you for pointing out these typos. We will correct them in the final version.

---

### Official Review · Reviewer_nR3E · 2023-11-01

**Soundness:** 3 good
**Presentation:** 3 good
**Contribution:** 4 excellent
**Rating:** 8
**Confidence:** 4

**Summary:**

The paper studies a fundamental problem of sampling from convex sets, the problem of sampling with respect to a log-concave distribution. Previous work [Mangoubi-Vishnoi, NeurIPS'22] has given an algorithm with $O^*(md)$ iteration and each iteration takes $O^*(md^{\omega-1})$ time. In this paper, the author improves the cost per iteration to $O^*(nnz(A)+d^2)$. This result directly answers the open problem proposed in [Lee-Sidford15, FOCS'15] which asks whether one can achieve such running time for the case $f\equiv 0$. To achieve this, the authors make use of the inverse maintenance technique in  [Lee-Sidford15, FOCS'15] and show one can compute the estimation of determinant to high accuracy by cleverly constructing an unbiased estimator and making use of the linear system solver as a primitive.

**Strengths:**

My general evaluation of this paper is very positive.  I did not manage to check the correctness of the proofs. Condition on the new Metropolis update rule and the log-determinant estimator is correct, I think everything goes through. This paper is technically solid and answers an important open problem.

**Weaknesses:**

I think the primary weakness of this paper is the presentation. I understand that due to the nature of theory papers, it's hard to present everything important within the page limits. The most interesting part of the paper  for me is how to get a good estimation of log-determinant and I think it deserves some space in the main paper.

**Questions:**

-

---

> ### Author Response · Authors · 2023-11-20
>
> Thank you for your valuable comments and suggestions. We are glad that you find that our paper is technically solid and answers an important open problem, and thank you for supporting our paper. We answer your specific question below.
>
> >The most interesting part of the paper for me is how to get a good estimation of log-determinant and I think it deserves some space in the main paper.
>
> Thank you for the suggestion. We discuss the estimation of the log-determinant on page 8 of the proof overview in Section 4. We would be happy to add a separate section that outlines the proof of Lemma B.3 (which allows us to show that our randomized estimator concentrates near its mean) in the main body of the paper.

---

### Official Review · Reviewer_6Phh · 2023-11-13

**Soundness:** 3 good
**Presentation:** 2 fair
**Contribution:** 3 good
**Rating:** 8
**Confidence:** 3

**Summary:**

This paper studies sampling from a logconcave distribution on a polytope. To this end, it uses a soft-threshold Dikin walk introduced in MV22. The paper signficantly improves upon the per iteration cost by applying the inverse maintenance techniques from LS15 and LLV20. The technical contribution is in showing how to apply these to the soft-threshold logbarrier.

**Strengths:**

I wasn't able to check the proofs, but the results suggest that the paper overcomes a technical difficulty prior works such as LS15, MV22, and LLV20 hadn't been able to address. Specifically, showing that the soft threshold logbarrier Hessian is slow-changing in a certain norm is a novel contribution of the paper, and it has consequences to improving the runtime of what is clearly an important problem.

--------

After rebuttal: I'm increasing my score and confidence.

**Weaknesses:**

I think the paper is, currently, not as self-contained as it should be. I believe this is easily fixable by adding relevant background material in the appendix. I also believe perhaps there may not be too much relevance of this paper in ICLR, and the paper would be much better appreciated in the theoretical CS community's conferences such as SODA, COLT, etc. Perhaps it would be helpful to state a more direct connection to ICLR.

**Questions:**

-

---

> ### Author Response · Authors · 2023-11-20
>
> Thank you for your valuable comments and suggestions.  We are glad you appreciate the novelty of our contribution and our improvements to the runtime of an important problem, and thank you for supporting our paper.  We answer your specific questions below.
>
> >I think the paper is, currently, not as self-contained as it should be. I believe this is easily fixable by adding relevant background material in the appendix.
>
> Thank you for the suggestion.  We will add additional background material to the appendix.  Specifically:  (i) We will include a more detailed discussion of the soft-threshold Dikin walk of [MV22] and of the regularized barrier function used in that walk.  (ii) We will include a more detailed discussion of the efficient inverse maintanance algorithm of [LS15].  (iii) We will also include additional background material on the implementation of the original Dikin walk given in [LLV20] for the special case when the target distribution is uniform on a polytope.
>
>
>
> >...the paper would be much better appreciated in the theoretical CS community's conferences such as SODA, COLT, etc.
>
> While the techniques in our paper would also be of interest to the theoretical CS community, we believe that our results on sampling from log-Lipschitz or log-smooth logconcave distributions constrained to polytopes have interesting applications to machine learning, including to training Bayesian machine learning models and to differential privacy. We discuss our improvements to the runtime over [MV22] for some of these applications in Section 2 and Table 1.
>
> We will include a section in the final version discussing runtime improvements in additional applications, including applications to differentially private convex empirical risk minimization for training privacy-preserving logistic regression and support vector machines models.

---

> > ### Comment · Reviewer_6Phh · 2023-11-22
> > **Thank you for your response!**
> >
> > Thank you for your response and for contextualizing your work (particularly in your responses to other reviews). Based on your responses, I'm happy to increase my score and confidence. All the best!

---

> > > ### Author Response · Authors · 2023-11-23
> > >
> > > Thank you for appreciating the contributions of our paper, and for increasing your score.  We are grateful for your helpful comments and suggestions.

---

### Meta-Review · Area_Chair_muM6 · 2023-12-06

**Metareview:**

This submission addresses the enhancement of sampling efficiency from log-concave distributions within polytopes. The authors present a nearly-optimal implementation of a Markov chain for log-concave sampling, showcasing a significant reduction in per-step complexity. This is achieved by introducing efficient linear solvers that leverage the slow change in matrices involved in the Dikin walk and by providing a novel approach to compute the determinantal term in the Metropolis filter step. The paper identifies and resolves technical issues in the [LLV20] algorithm and proof.

Most reviewers rated the paper highly for its technical contributions and practical relevance. One reviewer remained skeptical about its novelty, perceiving it as derivative of previous works. The authors actively engaged in the review process, comprehensively addressing concerns and demonstrating a commitment to improving their paper's accessibility and clarity. In the end, the paper's notable technical contributions make a strong case for its acceptance in ICLR 2024.

**Justification For Why Not Higher Score:**

Some concerns remain regarding the novelty of the work, in view of one reviewer's concerns.

**Justification For Why Not Lower Score:**

The other three reviewers feel that the paper effectively addresses significant technical issues in existing work, and offers tangible improvements in the efficiency of log-concave sampling.

---

### Decision · Program_Chairs · 2024-01-16

Accept (poster)